

# Crystallographic preferred orientations of ice deformed in direct-shear experiments at low temperatures

Chao Qi[1], David J. Prior[2], Lisa Craw[2], Sheng Fan[2], Maria-Gema Llorens[3], Albert Griera[3], Marianne Negrini[2], Paul D. Bons[4], and David L. Goldsby[1]

[1]Department of Earth and Environmental Science, University of Pennsylvania, Philadelphia, PA 19104, USA
[2]Department of Geology, University of Otago, Dunedin, New Zealand
[3]Departament de Geologia, Universitat Autónoma de Barcelona, Barcelona, Spain
[4]Department of Geosciences, Eberhard Karls University Tübingen, Tübingen, Germany

*Correspondence to:* Chao Qi (qichao1qc@gmail.com).

**Abstract.** We sheared synthetic polycrystalline ice at temperatures of $-5$, $-20$ and $-30°$C, to different shear strains, up to $\gamma = 2.6$ (equivalent strain of 1.5). Cryo-electron backscatter diffraction (EBSD) analysis shows that basal intra-crystalline slip planes become preferentially oriented parallel to the shear plane, in all experiments. This is visualized as a primary cluster of crystal $c$-axes (the $c$-axis is perpendicular to the basal plane) perpendicular to the shear plane. In all except the two highest-strain experiments at $-30°$C, a secondary cluster of $c$-axes is observed, at an angle to the primary cluster. With increasing strain, the primary $c$-axis cluster strengthens. With increasing temperature, both clusters strengthen. In the $-5°$C experiments, the angle between the two clusters reduces with strain. The $c$-axis clusters are elongated perpendicular to the shear direction. This elongation increases with increasing shear strain and with decreasing temperature. Highly curved grain boundaries are more prevalent in samples sheared at higher temperatures. At each temperature, the proportion of and irregularity of curved boundaries decreases with increasing shear strain. Subgrains are observed in all samples. Recrystallized grains and subgrains are similar in size and are both smaller than the original grains. Microstructural interpretations and comparisons of the data from experimentally sheared samples with numerical models suggest that the observed crystallographic orientation patterns result from a balance of the rates of lattice rotation (during dislocation creep) and growth of grains by strain-induced grain boundary migration (GBM). GBM is faster at higher temperatures and becomes less important as shear strain increases. These observations and interpretations provide a hypothesis to be tested in further experiments and using numerical models, with the ultimate goal of aiding the interpretation of crystallographic preferred orientations in naturally deformed ice.

## 1 Introduction

Polycrystalline ice 1h deformed in the laboratory (e.g., Kamb, 1972; Li et al., 2000; Wilson et al., 2014; Qi et al., 2017) and in nature (e.g., Gow and Williamson, 1976; Hudleston, 1977; Thorsteinsson et al., 1999; Treverrow et al., 2016; Weikusat



et al., 2017) develops strong crystallographic preferred orientations (CPO, often called crystal orientation fabric, COF, in the glaciological literature), usually presented as the preferred orientation of ice [0001] axes, i.e., $c$ axes. As single crystals of ice are most easily deformed by glide on the (0001) plane, i.e., the basal plane (Nakaya, 1958; Wakahama, 1967; Duval et al., 1983), the manner in which the $c$ axes are aligned affects the flow strength for a given applied deformation kinematics, for

example, simple shear versus uniaxial compression (e.g., Shoji and Langway, 1988; Azuma, 1995; Li et al., 1996; Duval et al., 2010; Budd et al., 2013). The development of a CPO is commonly used to explain the accelerating strain rate that occurs after secondary creep (Cuffey and Paterson, 2010, pp. 53-55). The occurrence of strong CPOs in ice has led to the use of enhancement factors (Jacka and Maccagnan, 1984; Budd and Jacka, 1989; Li et al., 1996) to relate laboratory flow laws for isotropic ice (Glen, 1955, 1975; Goldsby and Kohlstedt, 2001) to the flow of ice in glaciers and ice sheets where a strong CPO

is likely (e.g., Russell-Head and Budd, 1979; Thorsteinsson et al., 1999). Thus, understanding the formation and evolution of CPOs during deformation is crucial to our ability to predict rates of flow of ice sheets and glaciers as ice temperature rises in a warming world (Houghton, 1996) and as stress boundary conditions change, for example, during ice shelf thinning and collapse (e.g., Scambos et al., 2004; Joughin et al., 2014). Furthermore, the CPO controls elastic anisotropy in ice and, through this, the anisotropy of sound wave velocity (Kohnen and Gow, 1979; Diez et al., 2015; Vaughan et al., 2017). Seismic data

can be used to constrain bulk CPOs (e.g., Bentley, 1972; Smith et al., 2017; Kerch et al., 2018) and understanding how CPOs relate to deformation kinematics and conditions is valuable in limiting the range of possible CPO solutions in a field seismic experiment (Picotti et al., 2015; Vélez et al., 2016).

  Many laboratory experiments demonstrate the development of a CPO in polycrystalline ice. The vast majority of these experiments are on ice samples deformed in uniaxial compression to relatively small maximum strains (axial strains up to

~0.3) (e.g., Jacka and Maccagnan, 1984; Jacka and Li, 2000; Wilson et al., 2014; Qi et al., 2017; Vaughan et al., 2017). However, most deformation in ice sheets and glaciers is dominantly simple shear (Cuffey and Paterson, 2010) and CPOs developed in compression and simple shear are not equivalent (Alley, 1992). Shear experiments on polycrystalline ice are limited to relatively high temperatures and mostly relatively low shear strains (Kamb, 1972; Bouchez and Duval, 1982; Burg et al., 1986; Li et al., 2000; Wilson and Peternell, 2012). Only four published shear experiments (Bouchez and Duval, 1982;

Wilson and Peternell, 2012) achieve shear strains greater than 0.5 at temperatures $\leq -10^{\circ}$C, with the coldest experiment conducted at $-15^{\circ}$C. Shear experiments at colder conditions (Wilson and Peternell, 2012) have not achieved shear strains >0.12. The majority of laboratory shear experiments have been conducted at temperatures of $-5^{\circ}$C or warmer.

  In this contribution, we adapt the direct shear method, applied in rock deformation studies (e.g., Schmid et al., 1987; Dell'angelo and Tullis, 1989; Zhang and Karato, 1995; Heilbronner and Tullis, 2006; Kohlstedt and Holtzman, 2009) to poly-

crystalline ice. By confining our samples with gas pressure, we are able to apply relatively high differential stresses without causing brittle fracture of the samples, allowing us to shear ice to large strains at much lower temperatures than have been applied before. The objective of this paper is to explore the effects of shear strain and temperature on the CPOs and microstructures of ice deformed in shear, and to explore the implications for understanding the development of CPO in ice and the associated evolution of its mechanical behavior.



## 2   Methods

### 2.1   Sample preparation and deformation assembly

To prepare polycrystalline ice samples with a controlled initial microstructure, we adopted the flooding and freezing procedure developed by Durham et al. (1983) and Stern et al. (1997). Ice cubes made from deionized water were crushed in a blender into

ice powders. These powders were then sieved at $-30$ to $-20°$C to sizes between 0.18 and 0.25 mm for "standard-ice" samples (Durham et al., 1983). The sieved seed-ice grains were packed into stainless-steel cylindrical molds with an inside diameter of 25.4 mm and capped with double-o-ring-sealed stainless-steel end plugs, to a porosity of $\sim$40%. Air was then evacuated from the pores in the packed powder, while the molds containing the powders were equilibrated at $0°$C in an ice-water bath. Degassed, deionized water (at $0°$C) was then introduced into the evacuated powders in the molds. Flooded sample molds were

inserted into vertical holes in a large styrofoam block so that the mold bottoms rested on a copper plate at the bottom of a freezer maintained at $-30°$C. This ensured that the water in the saturated powders froze from the bottom up, excluding bubbles from the samples. After freezing overnight, ice samples were gently pressed out of the mold with an arbor press.

A 4 to 6 mm-thick slice was cut from the cylinders at $45°$ to the cylinder axis. Both surfaces were shaved to ensure they were flat and parallel to each other. These surfaces were then polished on 180-grit sandpaper. See Fig. 1(a) for a picture of a

prepared sample. The initial sample thickness, $h_0$, was measured with a micrometer. The ice sample was placed between two $45°$-cut aluminum or wood cylindrical pistons, as illustrated in Fig. 1(b). To prevent slippage between the ice and the pistons, 180-grit sandpaper was glued onto the cut surfaces of the pistons with epoxy. Ice samples and pistons were enclosed in an indium tube with an inner diameter of 25.4 mm, which was then encapsulated in another indium tube with an inner diameter of 26.9 mm, with the bottom of the tube "welded" to a 12.7 mm-thick stainless-steel spacer (here "welded" means the indium

tube was melted against the copper-plated steel spacer with a soldering iron). The total thickness of the two indium jackets is 1.4 mm. The bottom of a steel semi-internal force gauge was welded to the top of the outer indium jacket, with a 19.1 mm-thick zirconia spacer placed between the force gauge and sample to thermally isolate the ice from the welding area. During welding, the sample assembly was immersed in an alcohol bath at $\sim -60°$C, leaving only the small area to be welded above the bath.

### 2.2   Deformation Experiments

Samples were deformed in direct shear at a confining pressure $P = 20$ MPa at three different temperatures, $T = -5, -20$ and $-30°$C, in a cryogenic gas-medium apparatus (Durham et al., 1983; Heard et al., 1990), as illustrated in Fig. 1(c). Pressure and temperature were maintained constant to $\pm0.5$ MPa and $\pm0.5°$C, respectively. Before deformation, each sample was allowed to equilibrate at the deformation pressure and temperature for about 1 h. Deformation experiments were performed at a constant axial displacement rate of $5.08 \times 10^{-4}$ mm/s. Experiments were terminated at different shear strains. After each run, the

indium jackets were carefully peeled off, and then the deformed sample was photographed and placed into a long-term storage dewar filled with liquid nitrogen. The final sample thickness, $h_1$, was estimated from a photo, and we noted whether the sample thickness was constant along the length of the sample. The horizontal shift of the edge of the sample, synthetic to the imposed shear direction, $l$, was measured with a ruler (see Fig. 1(f)).





## 2.3 Data processing

The raw data, i.e., time, axial displacement and load, were processed to obtain shear strain, shear strain rate and stress data. In our experiments, assuming samples were deformed in simple shear, i.e., no flattening of the samples occurred and there was no deformation of the pistons, the calculated shear strain rate, $\dot{\gamma}_{\text{calc}}$, is given by $\dot{\gamma}_{\text{calc}} = \frac{\sqrt{2}v_{\text{ax}}}{h_0}$, where $v_{\text{ax}}$ is the axial displacement rate. The calculated shear strain is given by $\gamma_{\text{calc}} = \dot{\gamma}_{\text{calc}} \times t_t$, where $t_t$ is the total elapsed time of deformation. However, since samples shortened slightly in the axial direction and the wooden pistons were also slightly deformed, the calculated shear strains and shear strain rates sometimes deviated from the actual values for the samples. Moreover, some samples exhibit lateral bulging (Fig. 1(g)), the measurement of which provides an estimate of the degree of axial shortening, $\varepsilon_{\text{axial}}$. Thus, we measured the shear strain directly from the deformed samples and determined the shear strain rate from the measured shear strain. The shear strain of each sample is determined from the displacement measured on the sample by

$$\gamma_{\text{meas}} = \frac{\sqrt{2}l}{h_0}. \tag{1}$$

The measured strain rate of each sample is given by,

$$\dot{\gamma}_{\text{meas}} = \frac{\gamma_{\text{meas}}}{t_t}. \tag{2}$$

For the samples with $\gamma_{\text{meas}} \gg \varepsilon_{\text{axial}}$, the values of $\gamma_{\text{meas}}$ and $\gamma_{\text{calc}}$ are very close in value.

Shear stress is calculated from the measured axial load, $F_{\text{ax}}$,

$$\tau_{\text{raw}} = \frac{1/\sqrt{2}F_{\text{ax}}}{A_{s0}}, \tag{3}$$

where $A_{s0} = \frac{\sqrt{2}}{4}\pi \times 25.4^2$ mm$^2$ is the initial area of the shear surface. The strength of the indium jackets at a given temperature and strain rate, based on an indium flow law (W. Durham, personal communication), was subtracted from $\tau_{\text{raw}}$,

$$\tau_{\text{cor}} = \tau_{\text{raw}} - \tau_{\text{in}} \tag{4}$$

As strain increases, the area of the shear surface in contact with both top and bottom pistons decreases. A correction based on an autocorrelation function (Heilbronner, 2002) of the contact area, $A_{\text{acf}}$, can be applied to the shear stress,

$$\tau_{\text{acf}} = \tau_{\text{cor}} \frac{A_{s0}}{A_{\text{acf}}}. \tag{5}$$

Note that the shear stress and shear strain rate in our experiments have the following relationship with the von Mises equivalent stress, $\sigma$, and von Mises equivalent strain rate, $\dot{\varepsilon}$

$$\tau = \frac{\sigma}{\sqrt{3}}, \ \ \dot{\gamma} = \sqrt{3}\,\dot{\varepsilon}. \tag{6}$$

As illustrated in Fig. 1(d) for a typical stress-strain curve, with increasing shear strain, shear stress increases to a peak at a strain of 0.05 to 0.15 (equivalent to an axial strain of 0.03 to 0.09) and then decreases. Data collected for the peak stress





are used to characterize the mechanical behavior of ice with an initial, isotropic microstructure. This is equivalent to using the minimum strain rate in creep tests. The magnitude of the stress drop, $\Delta\tau$, is the difference in stress between the peak stress and where the slope of the stress-strain curve approaches zero, as illustrated in Fig. 1(d). The stress data at higher shear strains, although corrected for the change in contact area with strain, are affected by non-coaxial alignment of the pistons and bending

moments on the internal force gauge, and are not as robust as the peak-stress data. Processed data are presented in Table 1.

### 2.4   Collection of orientation data with electron backscatter diffraction (EBSD)

Samples were prepared for EBSD analysis in a scanning-electron microscope (SEM) at the University of Otago, following the procedure described in Prior et al. (2015). During preparation, a sample was either kept in a cryogenic dewar (at $\leq -190°$C) or in an insulated transfer box (at $\leq -120°$C). The deformation pistons were carefully separated from the ice sample with a

razor blade. The sample was mounted on a copper ingot, with a surface parallel to the shear plane facing up (see Fig. 1(e)). Mounting, polishing and analysis were performed in the same way as reported previously (Qi et al., 2017). Crystallographic orientation maps of all samples were obtained from the shear plane, with step sizes summarized in Table 2.

### 2.5   Microstructural analyses

Orientation data obtained from diffraction data were processed using HKL Channel5 software, including removal of single mis-

indexed points, and assigning the average orientation of neighboring pixels to un-indexed points. Grains were constructed from processed orientation data using the MTEX toolbox (Bachmann et al., 2011). Grain boundaries were drawn where neighboring pixel misorientations exceeded $10°$. No extrapolation of orientation data was applied in MTEX, since the data were already processed by the HKL Channel5 software. Grain size was determined as the equivalent diameter of a circle with the area of each grain in cross section. In the analysis of the average grain size for a map, grains containing no more than 5 pixels or lying

on the edge of the map were excluded. Orientation distributions were generated from either the complete set of orientation data or a subset of data with one point per grain using the MTEX toolbox in MATLAB (Bachmann et al., 2010; Mainprice et al., 2015). The manner in which an orientation distribution was generated is specified in the figure captions.

To quantify the strength of the CPOs, both the J-index (Bunge, 1982) and the M-index (Skemer et al., 2005) were used. From uniformly-distributed orientations to a single-crystal orientation, the J-index, based on a calculated orientation distribution

function, increases from 0 to infinity, while the M-index, based on the distribution of random-pair misorientation axes, increases from 0 to 1.

Since the CPOs of sheared ice are often characterized by double clusters of $c$ axes (e.g., Kamb, 1972; Hudleston, 1977; Bouchez and Duval, 1982; Jackson, 1999), an angle $\varphi$ was used to quantify the relative orientation between the two clusters. We adopted the same approach as previously described by Bouchez and Duval (1982). As illustrated in Fig. 2(b), in the

stereonets, an angle from $-90°$ to $+90°$ was defined on the shear plane (green circle). At a given angle, two great circles with $10°$ between them (red circles) were drawn perpendicular to the shear plane. The number of data points that lie between these two great circles were counted. The normalized counts were then plotted as the frequency at this angle on a histogram. The angle $\varphi$ was defined as that between the two peaks on the histogram (Fig. 2(c)).



## 3 Results

### 3.1 Starting material

The starting materials were the same as described in Qi et al. (2017). The undeformed samples of standard ice have a homogeneous foam texture, with polygonal grains and straight grain boundaries. The mean grain size is ∼0.23 mm. The initial
crystallographic orientation is approximately random with an M-index of 0.0026. There is almost no intracrystalline distortion within the grains.

### 3.2 Mechanical data

As illustrated in Fig. 1(f) and (g), the deformed samples exhibit some flattening normal to the imposed shear direction. As listed in Table 1, the measured thicknesses for deformed samples are very similar to the initial values. No slip between pistons and
samples was observed. Based on these observations, the maximum flattening (axial strain) in our samples is ∼19% in sample PIL145 (Table 1). This amount of axial strain is small relative to the shear strain, and does not affect significantly the strain ellipsoid nor the passive rotation of material lines (Sanderson and Marchini, 1984).

Graphs of shear stress plotted against shear strain (hereafter "stress" and "strain" for brevity) are presented in Fig. 3. Key parameters (e.g., peak stress) extracted from the experimental data are presented in Table 1. In Fig. 3(a), the stress was not
corrected for the changing area of the sheared surface (see Section 2.3), while in Fig. 3(b), this correction rotates curves on the stress-strain plot counterclockwise. All the curves for experiments with aluminum pistons show a rapid stress rise to a peak stress at an approximate strain of $\gamma = 0.1\pm0.06$, followed by a steep drop to a more slowly changing stress with increasing strain. The curves for experiments with wooden pistons have different shapes around the peak stresses, and smaller drops following the peak stresses. Two of the experiments with wooden pistons (PIL82 and PIL91) have complicated double peaks
near the peak stresses. In all experiments except for PIL82 and PIL135, the stress slowly decreases with increasing strain, following the steep drop after the peak stress. In PIL82, the stress increases slowly after a sharp drop at $\gamma \approx 0.2$. In PIL135, with increasing strain, the stress increases slightly and plateaus at $0.8 < \gamma < 1.9$, decreases at $1.9 < \gamma < 2.5$, and increases suddenly at $\gamma > 2.5$, which corresponds with the 45°-cut piston touching the metal cylindrical sleeve in the bore of the pressure vessel. In PIL144, the stress continues to decrease until a strain of ∼1.2, and increases slowly thereafter. At a strain of ∼1.8,
there is an upward perturbation in stress followed by a decrease in stress. In both PIL135 and PIL144, the perturbation in stress at a strain of 1.8 to 1.9 is probably related to the changes in kinematics due to the changes in the assembly geometry with increasing strain. PIL144 is the one sample for which the final sample thickness is not uniform along its length.

### 3.3 Crystallographic preferred orientations

In this subsection, CPOs in samples deformed at different temperatures to different shear strains are described, as illustrated in
Fig. 4. The CPOs of −5 and −20°C samples are all characterized by two clusters of $c$ axes. The primary cluster (M1) is normal to the imposed shear plane at all strains. The secondary cluster (M2) lies in the profile plane antithetic to the imposed shear



direction. The CPOs of $-30°$C samples are characterized by one broad cluster of $c$ axes close to the normal to the imposed shear plane. In all samples, $c$-axis clusters are elongated in the direction sub-perpendicular to the shear direction. At the same temperature, the CPO strength characterized by J- and M-indexes generally increases with increasing strain, with the exception of sample PIL87, which has a relatively small number of grains in the data set.

### 3.3.1 $-5°$C series

The CPOs of $-5°$C samples are all characterized by two tight clusters of $c$ axes. The angle between the two clusters decreases with increasing strain, as presented in Fig. 4(d). The secondary cluster is weaker (exhibits a lower value of multiples of uniform density, MUD) than the primary cluster, except for sample PIL87, which has a small number of grains in the data set. The elongation of the clusters is clearer in the higher-strain samples. The secondary cluster is stronger (exhibits a higher MUD value) in the stereonets plotted with all orientation data than it is in the stereonets plotted with one point per grain. The secondary cluster generally weakens (has a lower MUD value) with increasing strain in the stereonets plotted with one point per grain.

The CPOs of all $-5°$C samples feature girdles of $\langle 11\bar{2}0 \rangle$ axes ($a$ axes) and $\langle 10\bar{1}0 \rangle$ axes (poles to $m$ planes) in the shear plane, and weaker girdles of $a$ axes and poles to $m$ planes normal to the secondary $c$-axis clusters. $a$ axes in the three samples with lower strains have dominant clusters normal to the shear direction in the shear plane. In the sample with the highest strain (PIL94, $\gamma = 1.5$), the dominant cluster of $a$ axes is parallel to the shear direction. The distributions of the poles to $m$ planes are similar to the distributions of $a$ axes in all samples, except in the sample with the lowest strain (PIL91, $\gamma = 0.62$), where there is an additional cluster subparallel to the shear direction.

### 3.3.2 $-20°$C series

The CPOs of $-20°$C samples (Fig. 4) are also characterized by two clusters of $c$ axes. The angle between the two clusters increases slightly with increasing strain. The secondary cluster is weaker (has a lower MUD value) than the primary cluster. The elongation of the clusters is clearer in the higher-strain sample. The secondary cluster is stronger (has a higher MUD value) in the stereonets plotted with all orientation data than it is in the stereonets plotted with one point per grain. The secondary cluster also weakens (obtains a lower MUD value) with increasing strain.

In both samples, the CPO features broad girdles of $a$ axes and poles to $m$ planes in the shear plane. The distributions of $a$ axes and poles to $m$ planes exhibit maximum intensities subparallel to the shear direction in the shear plane. $a$ axes and poles to $m$ planes are more tightly clustered in the higher-strain sample.

### 3.3.3 $-30°$C series

The CPOs of $-30°$C samples (Fig. 4) are characterized by one broad cluster of $c$ axes close to the normal to the imposed shear plane. In the lowest-strain sample, the primary cluster is $\sim 10°$ oblique to the shear-plane normal. This probably relates to this sample fracturing on a plane slightly oblique to the shear plane when removed from the deformation piston and being analyzed



on this oblique surface. This primary cluster is asymmetric with a large number of $c$ axes distributed broadly in the quadrant antithetic to the shear direction. In the samples with lower strains (PIL142 and PIL143), the elongation of the primary cluster is more extensive in the stereonets plotted with one point per grain. In the sample with the lowest strain (PIL143, $\gamma = 0.65$), a weak secondary cluster is observed close to the primary cluster. This secondary cluster is stronger than the primary cluster in the stereonet plotted with one point per grain.

The CPOs of all $-30°$C samples feature broad girdles of $a$ axes and poles to $m$ planes subparallel to the shear plane, and clusters of $a$ axes and poles to $m$ planes subparallel to the shear direction. In the sample with the lowest strain (PIL143, $\gamma = 0.65$), $a$ axes and poles to $m$ planes form a complex pattern and multi-maxima subparallel to the shear direction.

## 3.4 Microstructure

In this subsection, microstructures in samples deformed at different temperatures to different shear strains are described. For a given temperature, as strain increases, the fraction of lobate grain boundaries decreases and the fraction of straight grain boundaries increases. In all samples, the distributions of grain size are skewed, with a peak at finer grain sizes and a long tail extending to coarser grain sizes.

### 3.4.1 $-5°$C series

The microstructures of all samples deformed at $-5°$C (Fig. 5(a)) are characterized by lobate grain boundaries and irregular grain shapes. In the sample with the lowest strain (PIL91, $\gamma = 0.62$), the grain boundaries are highly lobate. Preferred orientations of grain shape are not well-developed. Intragranular distortion and subgrain boundaries are widely observed in all samples. In all samples, most grains have $c$ axes sub-perpendicular to the shear plane (grains with reddish colors). In the samples with lower strains (PIL91, 82 and 87), the distribution is based on a very limited number of grains, leading to uncertainties in the mean grain size. Peak grain sizes (the grain size at the peak frequency of the distribution) for all samples are between 40 and 80 $\mu$m.

### 3.4.2 $-20°$C series

The microstructures of both samples deformed at $-20°$C (Fig. 6(a)) are characterized by slightly curved grain boundaries. In both samples, preferred orientations of grain shapes are not well-developed. Intragranular distortion and subgrain boundaries are observed. In the sample with lower strain (PIL145, $\gamma = 1.1$), grains with basal planes oblique to the shear plane (with other than reddish color) are widely observed. In the sample with higher strain (PIL144, $\gamma = 2.2$), the map is dominated by grains with basal planes subparallel to the shear surface (grains with reddish colors). The mean grain size decreases slightly with increasing strain. Peak grain sizes for both samples are 80 $\mu$m.



### 3.4.3 $-30°$C series

The microstructures of all samples deformed at $-30°$C (Fig. 7(a)) are characterized by straight and slightly curved grain boundaries and polygonal grain shapes. In the two samples with lower strains (PIL143, $\gamma = 0.65$ and PIL142, $\gamma = 1.4$), preferred orientations of grain shapes are not well-developed. In the sample with the highest strain (PIL135, $\gamma = 2.6$), there
is a grain shape preferred orientation, with long axes subparallel to the shear direction. Intragranular distortion and subgrain boundaries are observed in all samples. The subgrains are developed preferentially in larger grains and the shapes and sizes of the subgrains are similar to those of the small grains. In the sample with the lowest strain (PIL143, $\gamma = 0.65$), intragranular distortion is more evident and more subgrains are observed than in the sample with the highest strain (PIL135, $\gamma = 2.6$). In all samples, there are a range of grain orientations as shown by multiple colors on the map. In the two samples with lower strains
(PIL143, $\gamma = 0.65$, and PIL142, $\gamma = 1.4$), the mean grain size is 78 $\mu$m, while in the sample with the highest strains (PIL135, $\gamma = 2.6$), the mean grain size is larger, 101 $\mu$m. Peak grain sizes for all samples are between 40 and 60 $\mu$m.

## 4 Discussion

### 4.1 Mechanical evolution

The stress drop following the peak stress is usually attributed to dynamic recrystallization and/or geometric softening. Dy-
namic recrystallization often results in a grain-size reduction that is thought to cause weakening by increasing the strain-rate contribution of grain-size sensitive deformation mechanisms (e.g., Tullis and Yund, 1985; De Bresser et al., 2001). Geometric softening due to the development of a CPO also causes weakening (e.g., Hansen et al., 2012), particularly in a strongly viscously anisotropic material such as ice. As is evident from the microstructures and CPOs of deformed samples, both dynamic recrystallization and geometric softening occur in the experiments. It is difficult to separate the effects of the two processes in
the type of experiments carried out in this study. Due to uncertainties in the evolution in the effective cross-sectional area in the experiments of our samples, we cannot reliably establish whether the stress increases, decreases, or remains constant after the dramatic decrease in stress after the peak stress.

### 4.2 The orientation of the two $c$-axis clusters: comparison of experiments

In our experiments, the primary cluster of $c$ axes is normal to the shear plane in all deformed samples. This statement is true for
most other ice samples deformed dominantly by simple shear in the laboratory, for which CPOs with significant numbers of measured grains are published (Kamb, 1972; Bouchez and Duval, 1982; Li et al., 2000). Wilson and Peternell (2012) reported the primary cluster being slightly oblique to the imposed shear plane, but the sample images (Fig. 6a in Wilson and Peternell, 2012) show that shear zones developed oblique to the imposed shear plane and the $c$-axis cluster is sub-perpendicular to the shear zone boundaries.
Both $c$-axis clusters in our experiments are elongated in the direction sub-perpendicular to the shear direction. This elongation has been observed in many previous studies (Kamb, 1972; Bouchez and Duval, 1982; Li et al., 2000; Wilson and Peternell,


2012). Li et al. (2000) attributed this elongation to the extensional deformation in the shear plane normal to the shear direction, due to the flattening of the sample during shear deformation.

The evolution of $\varphi$, the angle between the two $c$-axis clusters, with strain in our samples is compared with the results from previous experimental studies, with numerical models and with data from naturally deformed ice (Table 3 and Fig. 8). We

have used the method described in Section 2.5 to measure $\varphi$ for our own data and also to make comparable measurements using literature data. For data from the literature, we digitized $c$-axis orientations from published stereonets (Bouchez and Duval, 1982; Burg et al., 1986; Li et al., 2000; Wilson and Peternell, 2012; Hudleston, 1977; Jackson, 1999; Van der Veen and Whillans, 1994). The values of $\varphi$ for experimental samples of Kamb (1972) are taken from that paper (Kamb published only contoured data); these angles were analyzed using a similar method to ours.

To our knowledge, Fig. 8 contains data from all published CPOs from experiments where simple shear is the dominant deformation kinematic. The values of $\varphi$ are scattered between 30 and 80° for all experimental samples with double $c$-axis clusters. Single-cluster CPOs occur at shear strains above 1.4. Many individual data sets, including our data at $-30°$C and $-5°$C, reveal that $\varphi$ decreases with shear strain. In our data, we observe that the trajectory of $\varphi$ with strain occurs at lower $\varphi$ values at $-30°$C than at $-5°$C, with the $\varphi$ value of one of the $-20°$C data points lying between the $-30°$C and $-5°$C

trajectories. The high-strain $-20°$C sample has a geometry suggesting that it departed from simple shear in a different way than the other samples (i.e., it was wedge-shaped after deformation: Table 1) and the resulting high $\varphi$ value may be anomalous. The correlation of the position of the $\varphi$–strain trajectories and temperature is less clear in the broader literature data. There are a number of possible reasons. The most likely explanation is that the data in Fig. 8 represent experiments with subtly different kinematics (deviations from perfect simple shear) and contains experiments conducted across a range of strain rates (or

stresses). For example, the data of Burg et al. (1986), which have lower $\varphi$ values than samples deformed at colder temperatures to the same strain, are kinematically different from all samples in the other experiments and contain very small numbers of grains in each data set. Burg et al. (1986) conducted see-through deformation experiments, for which the deformation is constrained to be simple shear. In all the other experiments, the samples are allowed to change length in the direction normal to the shear plane (i.e., to have a component of flattening), although in all cases the longitudinal strains normal to the shear plane

are small relative to the shear strains. At present, there are not enough experimental data to restrict the data set to samples with identical kinematics and strain rates. We would predict that given identical kinematics and strain rates, the angle $\varphi$ between the two clusters would decrease with increasing strain at any given temperature and that the $\varphi$–strain trajectory would shift to lower $\varphi$ values with decreasing temperature.

Except for this study, a single-cluster distribution of $c$ axes in experimentally sheared ice was only observed in Li et al.

(2000). In their study, the CPO with a single $c$-axis cluster occurs at a strain similar to the strain at which this CPO occurs in our experiments, but at a much warmer temperature ($T = -2°$C). Li et al. (2000) attributed the occurrence of this CPO to allowing free deformation of the samples in the direction perpendicular to the applied shear direction (i.e., flattening of the sample). Wilson and Peternell (2012) did not report a CPO with a single $c$-axis cluster at $-2°$C, even though their experiments were conducted using the same apparatus and kinematic constraints as those of Li et al. (2000). The experiments of Li et al.

(2000) were conducted to higher shear strains than those of Wilson and Peternell (2012), and also to higher shear strains than





our highest-strain experiment at $-5°C$. Allowing free deformation of the samples in the direction perpendicular to the applied shear direction may be important for explaining the single $c$-axis cluster (as suggested by Li et al., 2000). Our view, based on our study, is that the key element in generating a single $c$-axis cluster is high shear strain.

### 4.3 The orientations of the two $c$-axis clusters: comparison with models

The pattern of $\varphi$-strain trajectories in Fig. 8 is complicated. Individual $\varphi$-strain trajectories at one temperature do not match very simple models, such as one based on the evolution of the angle between the long axis of the strain ellipse and the shear direction, or the passive rotation of a line originally perpendicular to the shear plane (Fig. 8). Bouchez and Duval (1982) applied the two-dimensional kinematic model of Etchecopar (1977). Although this model roughly fits their three experimental data points, the predictions mirror the passive rotation of a line originally perpendicular to the shear plane and do not match the broad set of

experimental data. Van der Veen and Whillans (1994) predict different $\varphi$-strain trajectories (Fig. 8) depending upon model parameters, most particularly how recrystallization is incorporated into the model. We think the balance between different recrystallization mechanisms may be critical to the manner in which the angle $\varphi$ evolves. Modern, fast Fourier transform viscoplastic (VPFFT) models of intracrystalline deformation by dislocation glide (Lebensohn, 2001; Lebensohn et al., 2008) generate remarkably similar intragranular microstructures to those measured in ice deformation experiments to low strain

(Grennerat et al., 2012; Montagnat et al., 2014; Piazolo et al., 2015), and would seem to be an excellent starting point for trying to understand the evolution of CPO and microstructure in ice during deformation. Llorens et al. (2016a, b, 2017) have coupled the full-field viscoplastic code to recrystallization codes within the ELLE modeling platform (Jessell et al., 2001) to predict microstructural and CPO evolution in ice to relatively high strains. The bulk CPOs produced by these models (Llorens et al., 2016a, b, 2017) and those produced by earlier viscoplastic self-consistent models (Castelnau et al., 1996, 1997)

predict a single $c$-axis cluster in shear. The cluster is not perpendicular to the shear plane, but instead lies in an orientation antithetic to the shear direction with an angle to the shear plane normal that reduces with increasing shear strain (see Fig. 5(a)-(d) in Llorens et al., 2017). Localization occurs in these models (see Fig. 10 in Llorens et al., 2017), and the CPO patterns extracted from the localized zones of high strain rate have double $c$-axis clusters at low strains with $\varphi$ reducing with increasing strain, ultimately generating a single $c$-axis cluster at high strain (see Fig. 5(i) in Llorens et al., 2017). As these CPOs match

experimental results much better than bulk CPOs, we have extracted a data set of CPOs from the high-strain rate zones of two end member simple shear models from Llorens et al. (2017) to compare with experimental data. The data are for deformation without recrystallization (VPFFT only) from model experiment SSH0, and for deformation with 25 steps of recrystallization (grain boundary migration), including recovery driven by a reduction of the intra-granular stored energy for each increment of deformation, from model experiment SSH25, in Llorens et al. (2017). A selection of the $c$-axis stereonets from these model

experiments in shown in Fig. 9; measured values of $\varphi$ as a function of strain are shown in Fig 8. The main cluster of $c$ axes from the high strain-rate localized zones (Fig. 9) is still oblique to the shear plane normal, although the angle of obliquity is much less than in the bulk CPO data from the same models (see Fig. 5(a) and (d) in Llorens et al., 2017). The $\varphi$-strain trajectories for these two end-member models bracket most of the experimental data and support the idea that the complexity in the pattern of $\varphi$-strain trajectories relates to the role of recrystallization.





Clearly more work is needed to integrate and reconcile the outcomes of laboratory experiments and numerical models. This is an important direction to pursue as, in general, we have more constraints from laboratory experiments than we do from naturally deformed samples to provide a quantitative test of models. Two key ideas arise from our work. Firstly, numerical models fail to predict a $c$-axis cluster that is always normal to the shear plane. A possible explanation is that the models do not

include a key process that can affect grain orientations; grain boundary sliding (GBS) is a candidate process, and nucleation and/or preferential growth of grains with suitable orientations is another. Secondly, only the high-strain-rate zones of FFT-based models match broadly the experimental CPOs. An important focus for future research is to explore the balance of processes that occur in the high-strain-rate zones of models, and to see if we can re-parameterize the models accordingly or explore processes that enable the high strain-rate zone CPOs to propagate through a larger volume of the sample.

**4.4   The orientations of the two $c$-axis clusters: comparison with natural samples**

Many natural CPOs characterized by a single $c$-axis cluster occur, and some of these are attributed to shear deformation. It is difficult for us to make any comparison between CPOs occurred in natural samples and experimental samples here, as many of the natural samples are not from areas with large-scale shear context (e.g., Treverrow et al., 2016). It is possible that these data represent CPOs in ice sheared to high shear strains. The key difference between these single-cluster CPOs in natural samples

and those generated in experiments is that the experimental samples all have an elongated $c$-axis cluster, whereas the naturally deformed samples mostly do not.

Three studies of naturally deformed ice provide more context, because the shear zone geometries are constrained from field data. Hudleston (1977) presented a well-documented study on CPOs observed in a glacial shear zone, which were compared with the CPOs observed in experimental samples by Bouchez and Duval (1982). The ice studied by Hudleston (1977) was col-

lected from a shaft in the Barnes Ice Cap where the temperature was nearly constant at $-10°$C, and where the shear kinematics and strains were constrained by classical methods of structural geology. Key elements of Hudleston's (1977) observations that match those from shear experiments are that (1) the primary cluster of $c$ axes is close to normal to the shear plane at all shear strains; (2) the angle between two clusters, $\varphi$, decreases with increasing strain; and (3) CPOs with a single $c$-axis cluster occur at high strains. The transition between CPOs of double clusters and a single cluster occurs at shear strains between 1.1 and 2.7

(Bouchez and Duval, 1982), in good agreement with experimental data (Fig. 8). The $\varphi$–strain trajectory of this data set also fits within the range of experimental observations.

The CPOs of ice within the marginal shear zone reported by Jackson (1999) exhibit a wider range of $\varphi$ than the CPOs of ice deformed in the laboratory, from a single $c$-axis cluster to double $c$-axis clusters with $\varphi \approx 90°$. Because measurements of shear strains are unavailable for these natural ice samples, the relationship between $\varphi$ and strain cannot be determined. These CPOs

are all likely to be from ice deformed to shear strains $> 3$. The occurrence of double clusters at these strains would not match the experimental data well and illustrates that there are probably significant complications in natural scenarios. Wilson and Peternell (2011) show CPOs with single and double $c$-axis clusters that they attributed to simple shear. The data of Wilson and Peternell (2011), however, cannot be related quantitatively to strain. One aspect that the data of Jackson (1999) and some of the data of Wilson and Peternell (2011) have in common with the experimental observations is that $c$-axis clusters are elongated.





In the case of the data of Jackson (1999), it is notable that the elongation is perpendicular to the shear direction, as it is in all experiments.

## 4.5 Recrystallization processes

After deformation, all samples have significantly-altered microstructures, indicative of dynamic recrystallization. The mean grain sizes and the peak grain sizes in deformed samples are smaller than the initial grain size of 230 $\mu$m (Qi et al., 2017), indicating that nucleation is involved in the recrystallization process. The observation of subgrains within larger grains with similar sizes to smaller recrystallized grains suggests that subgrain rotation recrystallization (polygonization) (e.g., Guillope and Poirier, 1979; Urai et al., 1986; Alley, 1992) is a possible nucleation mechanism.

At $-5°$C, the presence of highly curved and lobate grain boundaries suggests that the recrystallization process is dominated by strain-induced grain boundary migration (GBM) (Urai et al., 1986). As temperature decreases, the fraction of lobate grain boundaries decreases, suggesting that GBM contributes less to recrystallization, while the fraction of straight grain boundaries and polygonal grains increases, indicating that lattice rotation and subgrain rotation (polygonization) contribute comparatively more to recrystallization. At $-30°$C, the large number of small polygonal grains and subgrain boundaries indicates that the recrystallization process is dominated by subgrain rotation recrystallization.

The microstructures of the samples evolves with increasing strain. At all temperatures, the fraction of curved grain boundaries decreases, with increasing strain, while the number of polygonal grains increases. This observation suggests that the transition of the dominant recrystallization process from GBM to lattice rotation also occurs with increasing strain.

## 4.6 Process control on CPO development

The transition from a double-cluster to a single-cluster distribution of $c$ axes with increasing strain and/or decreasing temperature corresponds with the transition in the dominant recrystallization process. Qi et al. (2017) proposed that deformation by lattice rotation and recrystallization by grain boundary migration are the primary controls on CPO development. We extend this idea here to explain the CPOs that are formed in shear (Fig. 10). Fig. 10(c) explains how key processes (Fig. 10(a)) involved in deformation and recrystallization may effect CPO formation and evolution in shear.

The VPFFT models that yield the results shown in Fig. 9(a) simulate the effects of lattice rotation, with glide primarily on the basal plane. The VPFFT models predict that lattice rotation generates an initial CPO with $c$-axis clusters perpendicular to the shear plane and parallel to the shear direction. The cluster perpendicular to the shear plane strengthens rapidly with shear and migrates slowly in a direction antithetic to the shear-induced vorticity. The cluster parallel to the shear direction weakens and rotates rapidly in a direction synthetic to the shear-induced vorticity.

In the GBM process, grains with low dislocation density consume those with high dislocation density by migration of their mutual boundary (Urai et al., 1986). Grains poorly oriented for easy (basal) slip (i.e., that have low resolved shear stresses, or Schmid factors, on the basal plane) have to deform by slip on non-basal slip systems. As these dislocations are more difficult to glide, and there will be more than one interacting slip system, internal distortion (a proxy for dislocation density) tends to be higher in grains whose basal planes are in low-Schmid-factor orientations (Bestmann and Prior, 2003; Vaughan et al., 2017).





Grains in high-Schmid-factor orientations will grow at the expense of grains in low-Schmid-factor orientations (Fig. 10(a) and (c)). In simple shear, high-Schmid-factor orientations on the basal plane occur where $c$ axes are normal to the shear plane and parallel to the shear direction (Fig. 10(b)).

The subgrain rotation process is built into the VPFFT model (Fig. 9(a)). The process can be considered as kinematically
indistinguishable from the lattice rotation process. The subgrain rotation process can lead to generation of new small grains, a nucleation process commonly called subgrain rotation recrystallization (Guillope and Poirier, 1979; Urai et al., 1986; Bestmann and Prior, 2003). Microstructural studies of rocks show that small recrystallized grains have CPOs that are randomly-dispersed equivalents of the stronger host-grain CPOs (Jiang et al., 2000; Bestmann and Prior, 2003; Storey and Prior, 2005). These observations are interpreted as the result of an increased contribution of GBS to deformation. Recent experiments, in which very
coarse-grained ice is recrystallized (Craw et al., under review) during deformation at $-30°$C, also reveal CPOs in recrystallized grains (with peak grain sizes of 125-175$\mu$m) that are randomly dispersed equivalents of the stronger CPOs in host grains (several mm grain size). These observations suggest that GBS can be an important process of deformation in ice (Goldsby and Kohlstedt, 2001). GBS may add a component of rotation around the vorticity axis (Fig. 10(c)), synthetic to shear (Cross et al., 2017).

Fig. 10(d) provides a schematic synthesis of the relationships of CPO patterns to temperature and shear strain as suggested by experimental data. The figure also attempts to explain the patterns in terms of the contributions of the processes outlined in the previous paragraphs. Both lattice rotation and GBM will generate initial CPOs with $c$-axis clusters perpendicular to the shear plane and parallel to the shear direction. Lattice rotation is likely the primary cause of rotation of the secondary cluster towards the primary cluster with increasing shear strain. The elongation of clusters perpendicular to the shear direction
may relate to the deformation kinematics (Li et al., 2000), but may also relate to GBS-aided rotation around the vorticity axis synthetic to the shear direction. GBM will continue to favor growth of grains with $c$ axes perpendicular to the shear plane and parallel to the shear direction throughout the deformation. Lattice rotation ensures that there is always a supply of grains with $c$ axes perpendicular to the shear plane that can grow by GBM. Similarly, lattice rotation depletes the supply of grains with $c$ axes parallel to the shear direction. Subgrain rotation recrystallization and GBS will contribute to the formation of the CPO
by providing a wider range of crystal orientations, some of which can grow by GBM. If GBM is more effective than subgrain rotation recrystallization, re-population of grains with the secondary $c$-axis cluster subparallel to the shear direction will slow the rotation of the secondary cluster towards the primary cluster. As shear strain increases, the CPO becomes dominated by the primary $c$-axis cluster, so that there are fewer grains in low-Schmid-factor orientations. We suggest that this will reduce the number of grains with high dislocation density, effectively reducing the driving force for GBM. This would explain the
decrease in microstructures indicative of GBM as shear strain increases. We suggest that GBM activity reduces with increasing shear strain as a result of a reduced driving force for boundary migration. As temperature decreases, the mobility of grain boundaries decreases, and the contribution of GBM reduces relative to lattice rotation, with two effects: the rotation of the secondary cluster as a function of strain is more effective at lower temperatures, and the $c$-axis clusters are broader at colder temperatures.





### 4.7 Determination of deformation geometry in natural ice: measuring $a$-axis orientations

Ice CPOs with a vertical $c$-axis cluster are common in nature (Faria et al., 2014; Treverrow et al., 2016; Weikusat et al., 2017). Such CPOs could relate to vertical axial shortening or to shear with a horizontal shear plane. There is little intrinsic information in the $c$-axis distributions to enable distinction of these two interpretations. Elongated $c$-axis clusters and double $c$-axis clusters break the cylindrical symmetry expected for axial shortening (Wenk and Christie, 1991), and have symmetry consistent with shear. However, these are reported relatively rarely for natural ice samples (Jackson, 1999; Jackson and Kamb, 1997; Wilson and Peternell, 2011). Furthermore, there are good examples of symmetrical (not elongated) $c$-axis clusters in naturally deformed samples demonstrably related to shear (Hudleston, 1977). Elongation of $c$-axis clusters remains a bit of an enigma.

Our experiments show that the $a$-axis (and pole to $m$-plane) distributions are not uniformly distributed in the planes perpendicular to $c$-axis clusters. In contrast, the $a$-axis distributions corresponding to CPOs with a single $c$-axis cluster formed in axial-compression experiments (e.g., the sample shown in Fig. 11 in Prior et al., 2015) are uniformly distributed within the plane perpendicular to $c$-axis cluster (Prior, unpublished data). Thus the distributions of $a$ axes (and poles to $m$ planes) in naturally deformed samples may help resolve their deformation kinematics with clustered $a$ axes indicating shear. Furthermore, if collected natural samples are oriented, the $a$-axis data may constrain the shear direction. Our experiments suggest that at high shear strains, $a$ axes and poles to $m$ planes are oriented parallel to the shear direction.

### 4.8 Future directions

Fig. 10 provides a hypothesis to test in future laboratory experiments. Key developments are to generate a set of experiments that can be subdivided on the basis of temperature, strain rate (stress) and kinematics. Exploring the role of initial grain size and chemistry will also be important. Because it is possible to use the direct shear method with a confining pressure, a wider range of temperatures and strain rates can be explored than has been achieved before this study. The direct shear approach has significant promise in expanding our understanding of sheared ice. However, control of sample kinematics (and through this, the details of strain rate) may be improved by adapting other rock deformation approaches, such as confined torsion (Paterson and Olgaard, 2000; Pieri et al., 2001; Covey-Crump et al., 2016). Comparison with numerical models (Llorens et al., 2017) gave excellent insights into how different processes interact. Areas of poor comparison between experiments and numerical models highlight deficiencies in our quantitative understanding of ice deformation and recrystallization. Future work should explore the parameter space within models to maximize the agreement with experimental observations, and needs to focus on adding processes that are currently not included in models. GBS is a key process that should be incorporated into models. Many aspects of numerical models are limited in terms of dimensions and/or kinematics, and designing experiments that match these limitations is also important.

Ultimately, both experimental work and modeling needs to be linked to natural deformation. The relationships of CPO and strain, quantified in the well-constrained study of natural ice deformation by (Hudleston, 1977), match well with the experimental observations, giving us and previous authors (Bouchez and Duval, 1982) confidence that the results of laboratory



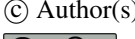

experiences are applicable to natural deformation. Studies comparable to Hudleston's are difficult undertakings. Overcoming these difficulties, so that we have more samples of naturally deformed ice across strain gradients, with constraints on deformation conditions, is crucial to future development of this research direction.

## 5 Conclusions

1. Confined, constant-displacement-rate, direct-shear tests allow laboratory deformation of ice approximating simple shear, to a shear strain ($\gamma$) of up to 2.6, at temperatures as low as $-30°$C. Samples, deformed at $-5$, $-20$ and $-30°$C, at shear strain rates of 8 to $20\times10^{-5}$ s$^{-1}$ show initial weakening over a shear strain interval of $\sim$0.3, after a peak stress at shear strains of 0.05-0.15. Strength evolves similarly for all samples deformed to higher strains, but whether the samples work harden or soften after the initial peak and stress drop cannot be established because of uncertainties in geometrical evolution. Samples were extracted for EBSD analysis of microstructures and CPOs.

2. Polycrystalline ice samples, sheared to different shear strains ($\sim$0.6 to 2.6) at $-5$, $-20$ and $-30°$C all develop a dominant, primary cluster of $c$ axes perpendicular to the shear plane. The orientation of this primary cluster does not change as a function of strain.

3. In samples deformed to $\gamma = 0.6$, 0.7, 1.4 and 1.5 at $-5°$C, $\gamma = 1.1$ and 2.2 at $-20°$C and $\gamma = 0.65$ at $-30°$C, a secondary $c$-axis cluster develops in the profile plane, but rotates from the primary cluster in a direction antithetic to the shear-related rotation. The angle between the two clusters reduces with shear strain in the $-5°$C experiments. Samples deformed to $\gamma = 1.4$ and 2.6 at $-30°$C exhibit a single $c$-axis cluster.

4. Clusters of $a$ axes and poles to the $m$ plane form, parallel to each other, within great circles perpendicular to $c$-axis clusters. At $-5°$C, these clusters lie roughly in the shear plane and are perpendicular and parallel to the shear direction, becoming parallel to the shear direction at the highest strain ($\gamma = 1.4$). At $-20$ and $-30°$C, these clusters lie parallel to the shear direction.

5. With decreasing temperature, both $c$-axis clusters become more diffuse, and the distinction of two $c$-axis clusters becomes less clear. At all temperatures, cluster strength increases with increasing shear strain. $c$-axis clusters are elongated along great circles perpendicular to the shear direction. Elongation increases with increasing shear strain.

6. Lobate grain boundaries are more prevalent and more irregular in samples sheared at higher temperatures. At each temperature, the proportion of and irregularity of lobate boundaries decreases with increasing shear strain. At all strains, the majority of grains are substantially smaller than the starting grain size. Subgrains occur in all samples at all temperatures. Subgrains are similar in size to recrystallized grains.

7. We used our data, published literature data, and comparisons of both with numerical models to interpret key processes that control the microstructures and the CPOs of ice during shear. We suggest that observed patterns result from a balance



of the rates of lattice rotation due to dislocation slip and growth of grains in high-Schmid-factor orientations by strain-induced GBM. GBM is faster at higher temperatures and becomes less important as shear strain increases. The data suggest that subgrain rotation is the most likely grain nucleation mechanism at all temperatures.

*Acknowledgements.* We are thankful to Jennifer Anderson and Travis Hager for their assistance with the experiments at University of Penn-

5   sylvania and Pat Langhorne for providing cold room facility at University of Otago. This work was supported by NASA fund (NNX15AM69G) and two Marsden Funds of the Royal Society of New Zealand (UOO1116 and UOO052). Collaboration to incorporate modeling work was supported by the Matariki fund of the University of Otago. LC and SF were supported by the University of Otago scholarships. LC was also supported by an Antarctica New Zealand scholarship.





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




**Table 1.** Summary of experimental data.

| Sample # | $h_0$ (mm) | $h_1$[1] (mm) | piston type[2] | $T$ (°C) | max $\varepsilon_{axial}$[3] | $\dot{\gamma}_{meas}$ ($10^{-5}$ s$^{-1}$) | $\gamma_{meas}$ | $\tau_p$[4] (MPa) | $\gamma_p$ | $\Delta\tau$[5] (MPa) | $\gamma_{\Delta\tau}$ |
|---|---|---|---|---|---|---|---|---|---|---|---|
| PIL82 | 6.35 | n/a | w | −5.4 | 0.15 | 7.98 | 0.69 | 0.87 | 0.07 | n/a[6] | n/a |
| PIL87 | 5.33 | n/a | w | −5.6 | 0.15 | 10.8 | 1.4 | 1.40 | 0.13 | n/a | n/a |
| PIL91 | 6.10 | ~6.4 | w | −5.6 | 0.10 | 17.8 | 0.62 | 1.54 | 0.09 | n/a | n/a |
| PIL94 | 5.69 | ~5.6 | Al | −5.2 | 0.14 | 13.7 | 1.5 | 2.02 | 0.10 | 0.73 | 0.31 |
| PIL135 | 5.33 | ~4.8 | Al | −30.5 | n/a | 12.5 | 2.6 | 7.03 | 0.16 | 1.43 | 0.38 |
| PIL142 | 5.08 | ~4.9 | Al | −29.9 | n/a | 12.8 | 1.4 | 6.73 | 0.16 | 1.32 | 0.25 |
| PIL143 | 5.08 | ~5.0 | Al | −30.6 | n/a | 14.6 | 0.65 | 6.81 | 0.10 | 1.32 | 0.26 |
| PIL144 | 5.33 | ~5.6[7] | Al | −20.4 | 0.15 | 11.9 | 2.2 | 4.60 | 0.13 | 1.50 | 0.38 |
| PIL145 | 6.10 | ~6.1 | Al | −20.1 | 0.19 | 9.49 | 1.1 | 4.80 | 0.05 | 1.91 | 0.35 |

[1] $h_1$ is the estimated value of sample thickness from photos of each sample. This estimation is only accurate to within ~10% of the value, due to the perspective and distortion in photos.

[2] w stands for wooden pistons. Al stands for aluminum pistons.

[3] max $\varepsilon_{axial}$ is the maximum axial strain estimated from bulging of the samples.

[4] $\tau_p$ is peak stress corrected for the strength of the indium jacket. $\gamma_p$ is the strain at which the peak stress was collected.

[5] $\Delta\tau$ is the magnitude of the stress drop following the peak stress. $\gamma_{\Delta\tau}$ is the approximate shear strain at which the sharp decrease in stress stops and stress becomes nominally constant.

[6] n/a stands for data not available.

[7] Sample has non-uniform thickness after deformation, i.e., is wedge-shaped. All other samples have uniform thicknesses.



**Table 2.** Summary of EBSD analyses.

| Sample # | data for CPO | | | data for microstructure | | |
|---|---|---|---|---|---|---|
| | section & step size ($\mu$m) | # indexed | # grains | section & step size ($\mu$m) | # indexed | # grains |
| PIL82 | shear, 50 | 36522 | 3303 | shear, 5 | 62636 | 84 |
| PIL87 | shear, 30 | 47261 | 1387 | shear, 6 | 236617 | 259 |
| PIL91 | shear, 50 | 113899 | 10560 | shear, 5 | 79486 | 225 |
| PIL94 | shear, 50 | 109767 | 8613 | shear, 7 | 335313 | 1186 |
| PIL135 | shear, 20 | 732565 | 33828 | shear, 5 | 454424 | 1282 |
| PIL142 | shear, 20 | 1131010 | 51529 | shear, 5 | 425804 | 2016 |
| PIL143 | shear, 20 | 344083 | 18061 | shear, 5 | 233143 | 1100 |
| PIL144 | shear, 30 | 416546 | 27738 | shear, 10 | 317027 | 4045 |
| PIL145 | shear, 30 | 312157 | 21248 | shear, 10 | 299413 | 3199 |





**Table 3.** The CPOs of sheared ice reported in the literature.

| Reference | sample name | type[1] | # of $c$ axes | $\varphi$ | conditions |
|---|---|---|---|---|---|
| Jackson (1999) | Chaos-1 | n | 72 | 84° | Camp Chaos marginal shear zone, 300-m depth |
| | Chaos-2 | n | 139 | 57° | |
| | Chaos-3 | n | 85 | 58° | |
| | Chaos-4 | n | 43 | 56° | |
| | Lostlove-1 | n | 59 | 46° | Lost Love marginal shear zone, 300-m depth |
| | Dragonpad-1 | n | 54 | single cluster | Dragon Pad marginal shear zone, 300-m depth |
| | Staging-1 | n | 87 | 42° | Staging area, 300-m depth |
| | Unicorn-1 | n | 22 | 51° | Unicorn Camp |
| | Fishhook-1 | n | 78 | 89° | Fishhook drill site |
| Hudleston (1977) | IV | n | 80 | 44° | Shear zone in the Barnes Ice Cap, lower strain part of core |
| | V-IX | n | 72-79 | single cluster | Shear zone in the Barnes Ice Cap, higher strain part of core |
| | X | n | 88 | 80° | Shear zone in the Barnes Ice Cap, lower strain part of core |
| | XI | n | 93 | 40° | Shear zone in the Barnes Ice Cap, lower strain part of core |
| | XII | n | 127 | 54° | Shear zone in the Barnes Ice Cap, lower strain part of core |
| Kamb (1972) | A10 | e | 462 | 79° | $T = -1.2°C$, $\dot{\gamma} = 0.2 \times 10^{-7}$ s$^{-1}$, $\gamma = 0.13$, $\varepsilon_{\text{axial}} = 0.13$ |
| | A8 | e | 507 | 65° | $T = -1$ to $-2.5°C$, $\dot{\gamma} = 2.8 \times 10^{-7}$ s$^{-1}$, $\gamma = 0.12\text{-}0.24$ |
| | A7 | e | 97 | 70° | $T = -2.5°C$, $\dot{\gamma} = 3.1 \times 10^{-7}$ s$^{-1}$, $\gamma = 0.05$, $\varepsilon_{\text{axial}} = 0.003$ |
| | A6 | e | 252 | 63° | $T = -4°C$, $\dot{\gamma} = 1.1 \times 10^{-7}$ s$^{-1}$, $\gamma = 0.40$, $\varepsilon_{\text{axial}} = 0.01$ |
| | A5 | e | 626 | 62° | $T = -2.5°C$, $\dot{\gamma} = 0.7 \times 10^{-7}$ s$^{-1}$, $\gamma = 0.27$, $\varepsilon_{\text{axial}} = 0.06$ |
| | A3 | e | 136 | 68° | $T = -4°C$, $\dot{\gamma} = 1.4 \times 10^{-7}$ s$^{-1}$, $\gamma = 0.16$, $\varepsilon_{\text{axial}} = 0.05$ |
| | A14 | e | 563 | 72° | $T = -1.2°C$, $\dot{\gamma} = 3.4 \times 10^{-7}$ s$^{-1}$, $\gamma = 0.15$, $\varepsilon_{\text{axial}} = 0.082$ |
| | A13 | e | 576 | 54° | $T = -1°C$, $\dot{\gamma} = 8.9 \times 10^{-7}$ s$^{-1}$, $\gamma = 0.22$, $\varepsilon_{\text{axial}} = 0.42$ |
| Bouchez and Duval (1982) | E1 | e | 147 | 60° | $T = -7°C$, $\dot{\gamma} = 1 \times 10^{-7}$ s$^{-1}$, $\gamma = 0.6$ |
| | E2 | e | 208 | 69° | $T = -12°C$, $\dot{\gamma} = 1 \times 10^{-7}$ s$^{-1}$, $\gamma = 0.95$ |
| | E3 | e | 100 | 35° | $T = -10°C$, $\dot{\gamma} = 1 \times 10^{-7}$ s$^{-1}$, $\gamma = 2$ |
| | Etchecopar-a | m | | 60° | $\gamma = 0.7$ |
| | Etchecopar-b | m | | 50° | $\gamma = 1.07$ |
| | Etchecopar-c | m | | 40° | $\gamma = 1.43$ |
| | Etchecopar-d | m | | 30° | $\gamma = 2.6$ |
| | Etchecopar-e | m | | 20° | $\gamma = 2.9$ |
| Burg et al. (1986) | SS4 | e | 56 | 58° | $T = -4°C$, $\dot{\gamma} = 1.79 \times 10^{-6}$ s$^{-1}$, $\gamma = 0.62$ |
| | SS5 | e | 72 | 31° | $T = -5°C$, $\dot{\gamma} = 1.34 \times 10^{-6}$ s$^{-1}$, $\gamma = 0.9$ |
| | SS6 | e | 47 | 39° | $T = -5°C$, $\dot{\gamma} = 1.49 \times 10^{-6}$ s$^{-1}$, $\gamma = 0.75$ |
| | SS7 | e | 60 | 44° | $T = -1.3°C$, $\dot{\gamma} = 1.44 \times 10^{-6}$ s$^{-1}$, $\gamma = 0.97$ |
| | SS8 | e | 58 | 40° | $T = -1°C$, $\dot{\gamma} = 1.47 \times 10^{-6}$ s$^{-1}$, $\gamma = 0.78$ |
| Van der Veen and Whillans (1994) | model 1 | m | | 68° | $\gamma = \sqrt{3}\varepsilon = 0.87$ |
| | model 1 | m | | 68° | $\gamma = \sqrt{3}\varepsilon = 1.73$ |
| | model 2 | m | | 72° | $\gamma = \sqrt{3}\varepsilon = 0.52$ |
| | model 2 | m | | 58° | $\gamma = \sqrt{3}\varepsilon = 1.04$ |
| | model 2 | m | | single cluster | $\gamma = \sqrt{3}\varepsilon = 1.56$ |
| Li et al. (2000) | A2 | e | | single cluster | $T = -2°C$, $\tau = \sqrt{3/2} \times 0.2$ MPa $= 0.24$ MPa, $\gamma = 2.18$ |
| | A5 | e | | single cluster | $T = -2°C$, $\tau = \sqrt{3/2} \times 0.3$ MPa $= 0.37$ MPa, $\gamma = 2.10$ |
| | A6 | e | | single cluster | $T = -2°C$, $\tau = \sqrt{3/2} \times 0.4$ MPa $= 0.49$ MPa, $\gamma = 2.04$ |
| Wilson and Peternell (2012) | 2-66-center | e | 387 | 61° | $T = -15°C$, $\tau = 0.4$ MPa, $\gamma = 1.0$, $\varepsilon_{\text{axial}} = 0.05$ |
| | 2-64-center | e | 712 | 59° | $T = -10°C$, $\tau = 0.4$ MPa, $\gamma = 1.1$, $\varepsilon_{\text{axial}} = 0.18$ |
| | 2-41-zone2 | e | 274 | 64° | $T = -2°C$, $\tau = 0.4$ MPa, $\gamma = 1.1$, $\varepsilon_{\text{axial}} = 0.08$ |
| | 2-41-zone3 | e | 128 | 53° | $T = -2°C$, $\tau = 0.4$ MPa, $\gamma = 1.1$, $\varepsilon_{\text{axial}} = 0.08$ |
| | 2-52-zone2 | e | 103 | 50° | $T = -2°C$, $\tau = 0.4$ MPa, $\gamma = 0.95$, $\varepsilon_{\text{axial}} = 0.1$ |
| | 2-52-zone3 | e | 942 | 68° | $T = -2°C$, $\tau = 0.4$ MPa, $\gamma = 0.95$, $\varepsilon_{\text{axial}} = 0.1$ |
| Llorens et al. (2017) | SSH0 | m | | 66° | $\gamma = \sqrt{3}\varepsilon = 0.52$ |
| | | m | | 56° | $\gamma = \sqrt{3}\varepsilon = 1.04$ |
| | | m | | single cluster | $\gamma = \sqrt{3}\varepsilon = 2.08$ |

[1] n stands for natural samples. e stands for experimental samples. m stands for models.



**Figure 1.** Figures illustrating the experimental procedure. Orange arrows indicate steps. (a) Photo of a sample and aluminum pistons. (b) Schematic drawing of the sample assembly before deformation. (c) Schematic drawing of the pressure vessel and the deformation assembly. (d) Schematic plot of a typical shear stress-shear strain curve during an experiment. Peak stress, $\tau_p$, and stress drop, $\Delta\tau$, are marked on the curve. (e) Schematic drawing of the sample assembly after deformation. The two sub-drawings at the bottom denote the shear plane and the profile plane (in red). (f) and (g) Photos of deformed samples at different perspectives.





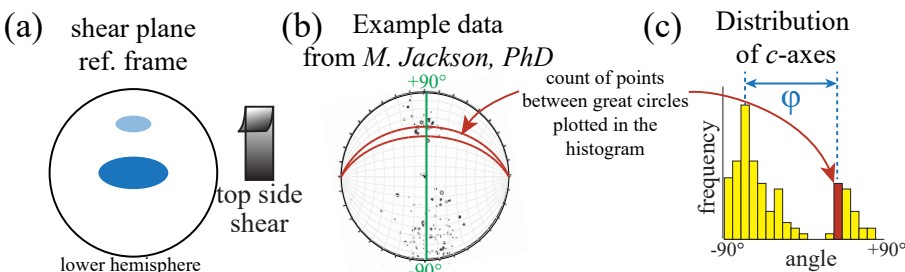

**Figure 2.** (a) Typical two-cluster distribution of $c$ axes on the stereonets in shear plane reference frame. (b) A schematic drawing explaining the method used to quantify the distribution of $c$ axes. (c) The distribution of $c$ axes plotted in a histogram, illustrating the angle between the two clusters of $c$ axes, $\varphi$.




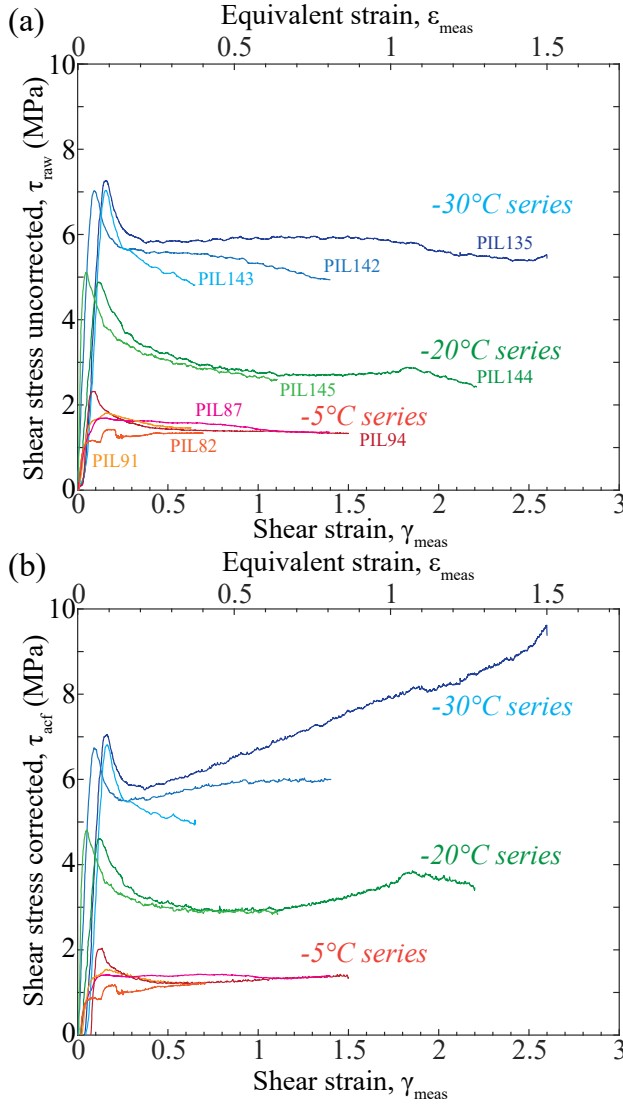

**Figure 3.** Plots of shear stress-shear strain curves for all experimental runs. (a) Shear stress, $\tau_{raw}$, calculated from the axial load. (b) Shear stress corrected for the strength of the indium jacket and the change in contact area, $\tau_{acf}$. Corrections for the strength of the indium jackets cause some curves to not pass through the origin.





**Figure 4.** Analysis of crystallographic orientations on the shear plane of all deformed samples. Samples are grouped by their experimental temperatures. Groups are separated by black horizontal lines. All data are from shear plane. The shear direction is top side up, as illustrated by the bold black arrow. Step size is between 30 and 50 $\mu$m (see Table 2 columns 2-4). J- and M-indexes are calculated based on all orientation data. (a) Distributions of orientations of [0001] axes from 1000 randomly-selected grains. (b) Distributions of orientations of [0001] axes contoured on the basis of one point per grain. (c) Distributions of orientations of [0001], [11$\bar{2}$0] and [10$\bar{1}$0] contoured on the basis of all orientation data. The contours on each stereonet are colored by multiples of a uniform distribution (MUD), values of which are indicated in the color bar next to the stereonet. (d) Distributions of the [0001] axes on the great circle normal to the shear surface. The angle between the two clusters, $\varphi$, is presented on each histogram.





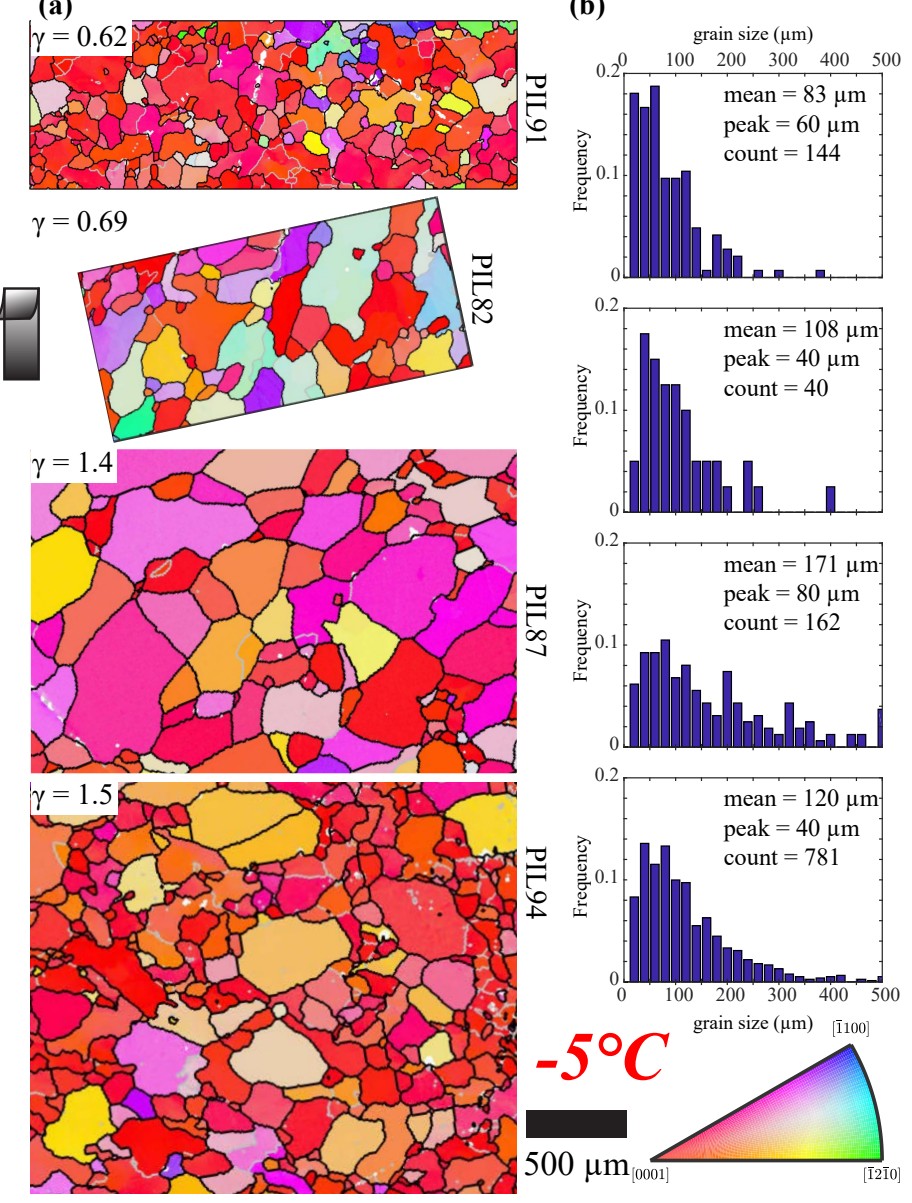

**Figure 5.** Microstructure at the shear planes of samples deformed at $-5°$. Strain increases from top to bottom. (a) Orientation maps colored by the crystallographic orientation normal to the shear surface according to the color map at bottom right. Step size is between 5 and 7 $\mu$m (see Table 2 columns 5-7). Grain boundaries, characterized by a misorientation of $10°$, are black, and sub-grain boundaries, characterized by a misorientation of $2°$, are gray. Un-indexed spots are white. The shear direction on the top side is up, as shown by the black arrow. (b) Distributions of grain size.





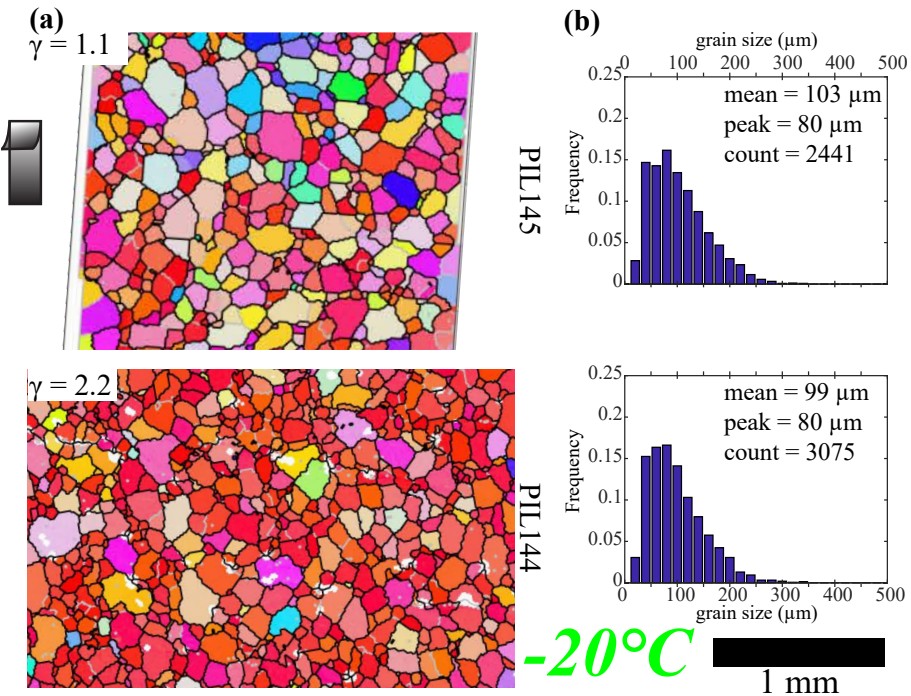

**Figure 6.** Microstructure at the shear planes of samples deformed at $-20°$. Strain increases from top to bottom. (a) Orientation maps colored by the crystallographic orientation normal to the shear plane according to the color map at bottom right. Step size is 10 $\mu$m (see Table 2, Columns 5-7). Grain boundaries, characterized by a misorientation of $10°$, are black, and sub-grain boundaries, characterized by a misorientation of $2°$, are gray. Un-indexed spots are white. The shear direction on the top side is up, as shown by the black arrow. (b) Distributions of grain size.



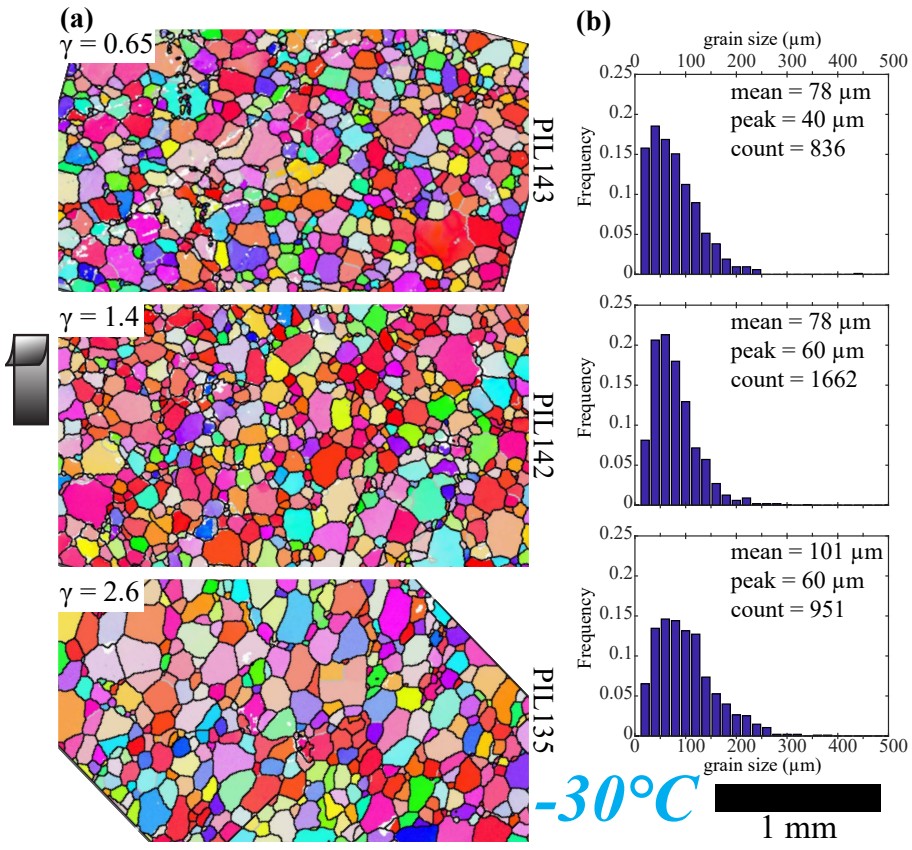

**Figure 7.** Microstructure at the shear planes of samples deformed at $-30°$. (a) - (c) Orientation maps colored by the crystallographic direction normal to the shear plane. Color map is the same as in Fig. 5. Step size is 5 $\mu$m (see Table 2 columns 5-7). Grain boundaries, characterized by a misorientation of $10°$, are black, and sub-grain boundaries, characterized by a misorientation of $2°$, are gray. Un-indexed spots are white. The shear direction on the top side is up, as shown by the black arrow. Strain increases from panel (a) to (c). (e) - (h) Distributions of grain size. From top to bottom, histograms are ordered the same way as the orientation maps.





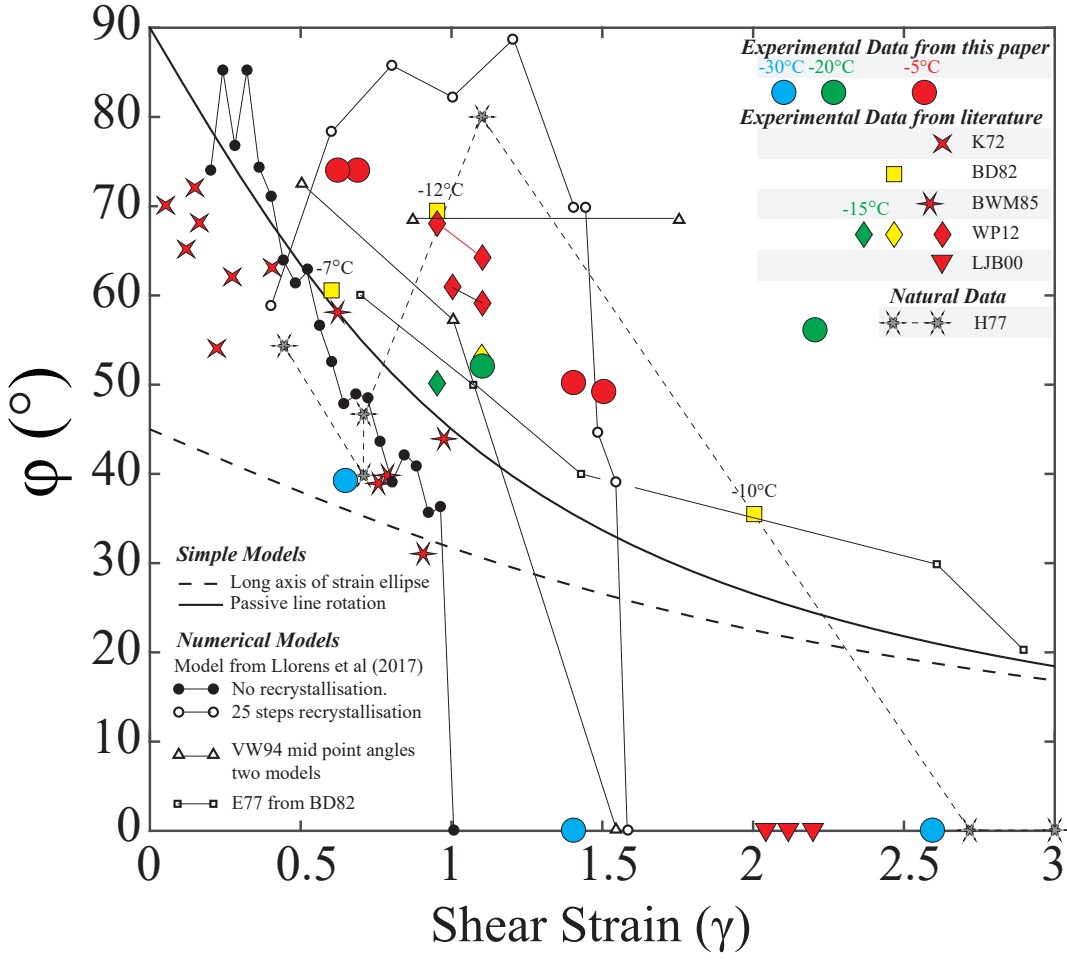

**Figure 8.** Plots of the angle between *c*-axis clusters, $\varphi$, for experiments from this study, experiments from the literature, data from natural shear zones, results of simple models, and results of numerical models. Experimental data: K72 = Kamb (1972); BD82 = Bouchez and Duval (1982); BWM86 = Burg et al. (1986); LJB00 = Li et al. (2000); WP12 = Wilson and Peternell (2012). Symbol colors broadly indicate deformation temperature and left-to-right position of symbols in legend indicates relative temperatures. Outcomes from published numerical models: VW94 = Van der Veen and Whillans (1994); E77 = the Etchecopar (1977) model as applied by Bouchez and Duval (1982). Models from Llorens et al. (2017) are explained in the text and CPOs from these are illustrated in Fig. 9. Data from natural shear zones: H77 = Hudleston (1977). Symbols tied by lines indicate they are from one sample.





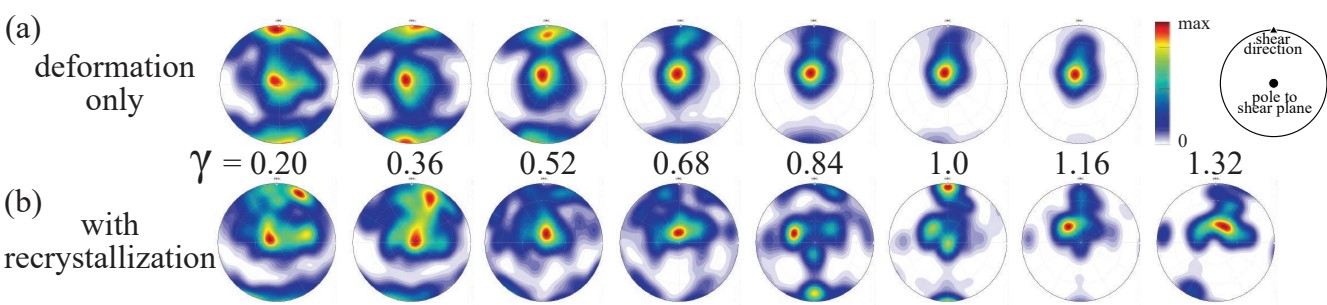

**Figure 9.** Lower-hemisphere, equal-area stereonets of $c$ axes from numerical models described by Llorens et al. (2017). The data are extracted from high-strain-rate domains in the models, and are presented as a function of shear strain. The color contours of each stereonet are normalized to its own maximum. (a) Model SSH0: deformation with no recrystallization. (b) Model SSH25: deformation with recrystallization (strain-induced grain boundary migration) and recovery.





**Figure 10.** Schematic drawing for the development of CPOs in sheared ice. (a) The evolution of microstructure. Four hexagonal ice grains with different initial orientations of basal planes are used to represent the microstructure. (b) Initial CPO and the kinematics in the shear surface. $\sigma_1$ and $\sigma_3$ are the maximum and minimum deviatoric stresses (compressive positive), respectively. (c) The effects from CPO-formation mechanisms. (d) The development of CPOs with strain at different temperatures. SGR = subgrain rotation. GBM = grain boundary migration. GBS = grain boundary sliding.