# Peer review of "Crystallographic preferred orientations of ice deformed in direct-shear experiments at low temperatures"

_The Cryosphere, 2018_

## Referee Comment (RC1) · M. Montagnat (Referee) · 25 Sep 2018

Dear authors.

I enjoyed reading your work, and I find it very complementary to some recent work that we did in Grenoble, and that my colleague Baptiste Journaux presented at AGU last year (publication is about to come...). In this work, we performed torsion test on ice polycrystals at high T, and followed the microstructure and texture evolution by EBSD. I will therefore mention it sometimes. Please consider my remarks below sometime as open questions, and sometimes as a participation to help improving some of your analyses or interpretations. The link between dynamic recrystallization mechanisms and

texture development remains an open question (even in the metallurgical community), and I do not pretend possessing the keys to its full understanding... Maurine

About the experimental procedure and results:

- What could be the impact of Piso on the recrystallization mechanisms? For instance, could it slow down the texture evolution, and the disappearance of the M2 pole? There exist very few work about it, I could just find the paper of Jones and Chew (J. Phys. Chem, 1983) that show the variation of the minimum creep rate with hydrostatic pressure... not sure it helps!

- M index is never used (or did I miss something?)...

- Could you expect any post-dynamic evolution of texture and grain size, especially in the case of PIL135 and PIL144? Therefore how caution should you be to use those results at the same level as the one from other experiments?

- You mention very often the elongation of the c-axis clustering. This elongation is not observed by Bouchez and Duval 1982 (later referred to as BD82), neither in natural ice from Hudleston (1977 or 2015 review paper), and not even in the deeper part of ice cores where shear is strong (as for instance bottom of Talos Dome ice core, Montagnat et al. 2012). We do not observe it neither in our recent torsion exp, going to gamma=2 (Journaux et al. AGU2017)... Therefore couldn't it result from the specific geometry of your test? It could be link to an extension component, cf Kamb 1972? The same remarks hold for the specific anisotropy of a-axis orientations on which I would be very careful before exploiting it too far in the conclusion...

- I am not sure to understand the interest of pole figures made by taking one point per grain. First because it does not include information about grain size, that is otherwise directly integrated, and second because it could induce a bias in case of a non normal distribution (and it is the case I think, regarding your microstructure). Then, since intragranular misorientations are strong, it can induce another bias depending on the

selected point inside the grain... I understand it in order to compare to "old" texture measurements that were done manually, but since the info is included into the "full" texture, not sure it is really interesting...

- in paragraph 2.5, about grain size measurement. How does it take into account the grain shape anisotropy? Such a method seems to be well adapted for a microstructure that evolves in staying "self-similar", but it is not the case at all, since the shape anisotropy (and the distribution of grain size) evolves with strain. Maybe you could just mention such a limitation? In the same idea, there is no mention of the fact that you are using a 2D technique to observe and evaluate grains that have a true 3D structures (with anisotropy). We all do that, of course, but the impact is different when a microstructure is relatively equiaxed, and remain so, or not. In particular, by doing so, we are totally unable to distinguish a small new grain from a cut piece of a highly serrated large grain... It therefore makes it complicated to distinguish nuclei (cf l. 13 of p.8). Again, you are using a "mean grain size", in the case of a non normal distribution, this mean has a weak meaning. Wouldn't a median + quartile representation be more adapted, in order, for instance to follow the evolution of the grain size distribution with strain?

- You refer to SGB, that you assume to be numerous and to evolve with strain. Honestly, this is hard to evaluate from the only figures 5 to 7, since there is no quantitative analyses of it. Relatively to some observations that I have done, I am even surprise not to see more of them, and I would be enable to say that there is a clear evolution with time. Couldn't you estimate, for instance, the Kernel Average Misorientation, and its evolution with time? (cf l.16 p 13 for instance).

About the discussion part:

- part 4.3 About the $\varphi$ angle, well, I have quite a lot to say (sorry...). First, does it really make sense to compare an evaluation of this angle performed based on very different materials, from experiments with very different conditions? In particular, if I want to

separate the two population of M1 and M2 orientation, I aim to take into account not only the visible angle between the 2 poles, but also the number of orientations on each pole (some kind of "weighted angle"). Why? Because in some cases there remain so few orientations in the M2 pole, that the separation is doubtful (and strongly dependent on the measurement technique). For instance, my interpretation of the gamma = 2 texture in BD82 is that there is no more M2 (or too few to be considered, and this was also the interpretation of BD82). Therefore, in this case, the phi angle has not meaning. I would do the same interpretation on your figure 2d at -20 and -30°C. Then, from my point of view, figure 8 is highly confusing. (a) you compare experiments made in drastically different conditions. For instance Burg et al. worked on thin plates of ice, with very few grains, and strong boundary conditions effects are expected. Kamb added some axial compression at different levels, Hudleston samples are from natural fault... (b) the weakness of the model used by Llorens et al. is also hidden in the figure. This model, by construction (because it requires non basal prismatic and pyramidal slip system to accommodate the deformation) is unable to stabilise the M1 pole to the vertical, as it is observed naturally or in the laboratory. Although the angle can be small between this pole and the vertical, it persists and the main reason is a non adapted representation of the mechanisms that are accommodating basal slip. Even when adding some representation of subgrain mechanisms can't they correct this bias. But when plotting only the phi angle, this problem is hidden, and the interpretation can be biased... Well, I am not sure that this phi angle evolution is so necessary to the interpretation of your results, and maybe this one would be clearer without trying to fit all other existing data???

- Part 4.5 Here, again is evoked the increase of subgrain boundary and lattice rotation effect without it being really quantified... It is therefore not so straighforward to assess an evolution from GBM to lattice rotation dominating process during recrystallization.

- Part 4.6 This paragraph is, to my point of view, giving a quite simplistic explanation for CPO development. It has been shown in ice and other materials, for quite some

time now, that stress and strain field heterogeneities prevent from making a clear distinction between grains "well oriented" and grain "badly oriented" (see e.g. Grennerat et al. 2012, Piazolo et al. 2015 for ice). Such a clear separation is holding when dealing with mean-field modeling approachs that are considering grain as an inclusion in a homogeneous equivalent medium. Full-field modeling approaches such as the one of Llorens et al. does reproduce the complexity. Furthermore, other mechanisms such as twinning (not in ice) and kink-banding (in ice) can be invoked to accommodate basal slip without the requirement of glide in non-favorable slip systems. And internal distortion does not always need non-basal dislocations to be formed... (tilt bands in ice are made of edge basal dislocations). And internal distortion is not a proxy of the dislocation density! It is only a proxy of the geometrically necessary dislocations, which are not necessarily correlated with the full dislocation density... Maybe some quantitative observation of a relation like subgrain density = f(schmid factor) could help? But we have tried, and we find no relation, such as the results of Grennerat et al. (2012).

- Part 4.7 seems highly speculative to me. Mainly because this could very likely result from this specific experimental set-up on which a compression or tension component may add to the shear deformation (see for comparison results of Duval 1981, JOG vol 27 who shows the effect of adding some compression on a shear experiment. Although they do not measure a-axis orientation, their resulting c-axis orientations could explain partly your a-axis distributions). Just to let you know, we do not observe this specific distribution in our recent torsion tests on ice (Journaux et al. AGU 2017). On more point on this paragraph, related to the references given for c-axis clusters observe in nature: it seems to me more fair to cite pioneer works when they exist (at least some) than to always refer to the review work or the more recent work.

- Conclusion:

From the remark about paragraph 4.7, I would suggest not to mention the point 4 of the conclusion, since it has not been shown that the observed results are not related to the specific experimental conditions used here. To turn this observation into a generic

tool to interpret natural texture appears to me a little too fast...

Similar remarks would hold also for the point 5, and the mentioning of the elongation of c-axis cluster that has not been observed in previous work (for instance Bouchez and Duval 1982).

About point 6, please refer to my comment on subgrain observations that are very weak in this paper. In order to be able to provide some info about subgrain size, one would need a measurement of this size, or at least a proxy, and it is not given in this work.

Point 7, a similar remark holds here too since, in order to link what the author calls "high Schmid-factor grains" to GBM, one would need to show that there exist a relation between the Schmid factor and the grain size, or grain shape for instance (if grain size or shape is taken as a proxy of GBM). And this is not provided here. And the last sentence appears speculative too, since, to be able to discriminate the nucleation mechanisms, one would need to be able to observe nuclei! But, because of the 2D observation tool used here, one cannot distinguish small new grains from cut part of a serrated grain boundary. On top of that, nucleation by bulging could very well occur, but one would not observe it by only looking at 2D microstructure at the end of the test. Therefore, the only qualitative observation of subgrain boundaries is not enough to discriminate the nucleation mechanism. So this could be your interpretation, or assumption, but to my point of view this is not shown by the results presented here.

―――――――――――――――――

---

## Referee Comment (RC2) · Anonymous Referee #2 · 25 Sep 2018

Summary

I have reviewed the manuscript, "Crystallographic preferred orientations of ice deformed in direct-shear experiments at low temperatures", submitted to The Cryosphere by Qi et al. In this study, the authors utilize a series of laboratory experiments and advanced materials characterization techniques to elucidate the microstructural properties of laboratory prepared specimens of polycrystalline ice under direct shear, taken to various strains at temperatures between -5°C and -30°C. In summary, the authors found through electron backscatter diffraction that the c-axis of the individual ice grains became clustered in an orientation perpendicular to the shear plane in all experiments,

with an additional c-axis clustering found at the lowest temperature and highest strain experiments. The authors go on to present a detailed discussion relating to the strains, temperatures, and stresses associated with the observed microstructures for each tested specimen and how they correlate to the various known modes and mechanisms of solid deformation. . .including dynamic recrystallization, strain induced grain boundary migration, and grain boundary sliding. These findings are thought to be of significance in that the easy-slip plane of ice Ih is along the basal plane perpendicular to the c-axis and because the resultant laboratory microstructures and that of natural ice are very comparable, such that the topic lends itself well to the intended scope of The Crysosphere.

Overall, I found the manuscript to be very well-written and the laboratory experiments and microstructural analysis presented robust. The conclusions, although perhaps a bit lengthy, were generally supported by the evidence shown throughout the manuscript and the discussion/interpretation of the results. Except for a few minor comments and questions for the authors, I recommend the timely publication of this manuscript in The Cryosphere and congratulate the authors on their work.

Recommendation: Minor Revisions

Comments/Questions

1. During the sample preparation, when the samples are cooled to -60°C for the welding of the indium jacket, is there any possibility for the thermal/confinement stresses to alter the microstructure as it would relate to the grown-in dislocation density?

2. From looking at Figure 1, I am perplexed as to how the piston is able to translate laterally while also remaining rigid and in-line with the axis of compression? Could you please explain?

3. Could you include the data (via personal communication) related to the flow law of the indium jacket and perhaps also the company/supplier that is used? Such that these

experiments could be repeated?

4. Please add a citation for Line 1-2, page 5, regarding using the minimum strain rate in creep tests.

5. In Section 3.4, please comment on the skewed distribution of grain sizes. Is this lognormal? As would be expected? Was this distribution used for calculating the mean? How was the anisotropy in the elongated grains accounted for?

6. Discussion Section. Although I appreciate the detail of this section, it seems to me that it could be more concise, such that the most relevant findings and results are more impactful.

7. In Section 4.1, should any consideration be given to the recrystallized grains experiencing primary creep in this scenario?

8. Page 11, line 30, replace "in" with "are"

9. Line 13-14, Page 13, please add a citation for this statement.

10. Regarding GBS mentioned in Section 4.6, was there any evidence of this in the observed microstructure? Quadruple points? If not, how would this be incorporated into models for ice if it has yet to be directly observed?

11. Conclusions Section. Could also be more concise. (e.g. no need to summarize the method and/or results before presenting a conclusion)

12. Figure 3b – and with regard to Question 2. . .Am I correct to interpret the increase in the shear stress in these tests as related to the piston becoming displaced and the onset of frictional effects? If not, could you further explain the cause of the increase in the shear stress?

13. Figures 5,6,7 – It's not clear to me what is being indicated with the blocky black arrow on the left of these maps. Is this a transverse view/map?

14. Figure 10 – Is it possible to also quantify the Key Processes related to the Final microstructure? Such that these 2-D characteristics could be identified with an automated algorithm? Perhaps see Lehto et al. 2016, Characterization of local grain size variation of welded structural steel, as a good starting point. It seems that there needs to be a better method of identifying and/or quantifying the differences in these microstructural regimes.

15. Lastly, after reading Maurine Montagnat's insightful comments pertaining to this manuscript, I would have to agree that it is difficult to ascertain with any certainty the nucleation mechanisms responsible for recrystallization from a 2-D post-mortem analysis alone.
* * *

---

## Author Comment (AC1) · 13 Nov 2018

Response to Reviewer 1 We thank Reviewer 1 for their thoughtful and helpful review of our paper. The Reviewer's comments are shown in blue, followed by our detailed responses, shown in black text.

1. What could be the impact of Piso on the recrystallization mechanisms? For instance, could it slow down the texture evolution, and the disappearance of the M2 pole? There exist very few work about it, I could just find the paper of Jones and Chew (J. Phys. Chem, 1983) that show the variation of the minimum creep rate with hydrostatic pressure... not sure it helps!

[Figure]

There is some work on the effect of pressure (P) on creep behavior. Cole (1996) and Rigsby (1958) show there is very little effect of pressure on dislocation glide in single crystals, and Durham et al. (1983) show little effect of pressure on tertiary creep of polycrystalline ice. We have just completed some repeat experiments of the -10°C experiments in Qi et al. (2017), but at a confining pressure of 20MPa (the Qi et al. (2017) experiments are conducted for a confining pressure of 10MPa). The peak stresses and shape of the stress-strain curve through tertiary creep are very similar at the two different pressures. Jones and Chew (1983) showed a change in minimum strain rate related to changes in pressure; it is difficult to evaluate this reported effect, however, as insufficient details are provided in the paper about how the load was applied and how the load was adjusted to give equivalent differential stresses for all experiments. Indeed, in all previous experiments, investigating the effect of pressure on deformation, except those in Durham et al. (1983) and in our own experiments, the axial stress is measured external to the pressure vessel and is likely to be subject to significant uncertainties, including the friction of the deformation piston which must pass through a sliding pressure seal. The measured stress in those studies thus may not be the stress on the sample, and since the stress exponent is relatively large (3 to 4), errors in stress measurement will significantly influence the measured strain rates. The wide range of experiments carried out by Bill Durham from the early 1980s to the 2010s (and now by us, using the same apparatus), where the load cell is located inside the pressure medium and unaffected by seal friction, do not show any significant effect of pressure on creep at low pressures (Durham pers comm). At high pressures, and more specifically at high differential stresses (∼>50MPa), the style of deformation changes (Golding, Durham and Prior in prep) from homogenous deformation to localized ductile shear (Golding et al., 2010, 2012). That is not relevant to the experiments presented here, nor to natural terrestrial ice deformation where P would rarely exceed 50 MPa and high differential stresses are only found near the margins of ice sheets, where P is low (Bons et al. 2018).

We do not think including an extensive discussion of possible pressure effects on deformation in the paper is valuable since all experiments were conducted at the same confining pressure. We have included a statement in section 4.2: "All previous shear experiments have been at ambient pressure. Durham et al. (1983) show that there is minimal effect of confining pressure on the tertiary creep of ice."

2. M index is never used (or did I miss something?)...

The J- and M-indexes are illustrated in Figure 4 for all samples.

3. Could you expect any post-dynamic evolution of texture and grain size, especially in the case of PIL135 and PIL144? Therefore how caution should you be to use those results at the same level as the one from other experiments?

After each experiment run, retracting the deformation piston and depressurizing the pressure vessel usually took a couple of minutes. The sample was then removed from the apparatus and transferred to the -30°C freezer. The indium jacket was peeled off of the sample in the freezer, which took another couple of minutes. The length and width of the samples were measured in the freezer. The samples were briefly exposed to room temperature air for about 30 s while they were photographed, then returned to the freezer. Samples were then quenched in liquid nitrogen by lowering them slowly into a liquid nitrogen storage dewar. The samples remained in liquid nitrogen until they were removed for EBSD analyses. The maximum total time from the end of an experiment to the time the sample was quenched in liquid nitrogen is ∼ 15 minutes.

Static annealing of the microstructures after the sample is unloaded is always a potential issue in deformation experiments. Repeat EBSD measurements of deformed ice samples after fixed thermal annealing times (Hidas et al., 2017) show some microstructural changes, but the changes that occur in one hour (or less) at relatively warm temperatures (-2°C, -5°C) are limited to relatively minor static recovery. Thus, the 15 mins our samples spend prior to quenching is unlikely to have affected the microstructures of our samples significantly. More specifically, the observed CPOs of our samples should represent those that existed at the end of deformation experiments.

[Figure]

Hidas et al. (2017) also show no significant grain growth in pre-deformed samples over the time scales of our sample extraction process. Experiments on samples with initially undeformed but very fine grains (∼10 miron initial size) show insignificant grain growth (Becroft, 2015) on the same time scales.

We have added a statement into section 2.2: "The maximum time between the end of an experiment and the sample being quenched in liquid nitrogen is ∼15minutes. Microstructural changes on this timescale are likely to be limited to minor static recovery (Hidas et al., 2017), with no significant change in CPO or distributions of grain size."

4. You mention very often the elongation of the c-axis clustering. This elongation is not observed by Bouchez and Duval 1982 (later referred to as BD82), neither in natural ice from Hudleston (1977 or 2015 review paper), and not even in the deeper part of ice cores where shear is strong (as for instance bottom of Talos Dome ice core, Montagnat et al. 2012). We do not observe it neither in our recent torsion exp, going to gamma=2 (Journaux et al. AGU2017)... Therefore couldn't it result from the specific geometry of your test? It could be link to an extension component, cf Kamb 1972? The same remarks hold for the specific anisotropy of a-axis orientations on which I would be very careful before exploiting it too far in the conclusion...

In Figure 5 of (Bouchez and Duval, 1982), samples E1 and E3 both shows an elongation in the c-axis clusters. Although these can be seen in the original data they are clearer when the data are reoriented so that pole to the shear plane is in the center of the stereonet. The re-plotting of the BD82 data that we used in our analysis is shown below. We have similar analyses for all the data we extract from the literature (from both experiments and natural ice) and it may be useful to show these as an extra resource. The c-axis cluster elongation is common to all the experiments where a large piece of ice is sheared, except one experiment in BD82 (E2) and one part of one experiment in Wilson and Peternell, 2012) (2-52 zone 2). Unpublished shear experiments from the Hobart lab all show elongated c-axis clusters (Adam Treverrow, pers. comm.), as do all samples from a new set of ∼ 20 experiments (conducted at temperatues of -7°C to
-20°C, for shear strains of 0.2 to 0.6) from John Platt and Tom Mitchell. (We recently conducted EBSD on the Platt and Mitchell samples at the University of Otago). The elongation in the c-axis clusters does not occur in the syn-microscopic experiments (Burg et al., 1986), but the kinematic boundary conditions of these experiments are very different, with a sample thickness that is less than the grain size. We are intrigued that the Reviewer does not observe elongation in their recent experiments. We look forward to seeing these data and discussing this issue with the Reviewer in more detail. In the meantime, in the paper, we factually state: "this elongation has been observed in many previous studies (Kamb, 1972; Bouchez and Duval, 1982; Li et al., 2000;Wilson and Peternell, 2012)".

The c-axis elongation may well be due to a component of sample elongation perpendicular to shear: we have already included this possibility in section 4.2: "Li et al. (2000) attributed this elongation to extensional deformation in the shear plane normal to the shear direction, due to the flattening of the sample during shear deformation." It is important to note that this flattening occurs in situations where there is zero normal stress perpendicular to the shear plane (Li et al., 2000) as well as when there is a compressional component normal to the shear plane (e.g. in our experiments). We are not totally convinced that this kinematic explanation is sufficient; the statement we make in section 4.7, that "elongation of c-axis clusters remains a bit of an enigma" reflects our current level of understanding. We agree that the c-axis cluster elongation we observed in our experiments is uncommon in naturally deformed ice samples. In section 4.4. we state: "The key difference between these single-cluster CPOs in natural samples and those generated in experiments is that the experimental samples all have an elongated c-axis cluster, whereas the naturally deformed samples mostly do not." In section 4.7 we state: "Furthermore, there are examples of symmetrical (not elongated) c-axis clusters in naturally deformed samples demonstrably related to shear (Hudleston, 1977)". To make this clearer we have added a statement into section 4.4, where the Hudleston data are discussed: "Hudleston (1977) did not observe elongated c-axis clusters. "

Since a-axis orientations have never been documented for sheared ice samples before, they are useful for future comparisons with naturally and experimentally deformed ice. We agree that it is not necessary to push the interpretation of these data too far at present, so we only described our observations in the conclusions section.

5. I am not sure to understand the interest of pole figures made by taking one point per grain. First because it does not include information about grain size, that is otherwise directly integrated, and second because it could induce a bias in case of a non normal distribution (and it is the case I think, regarding your microstructure). Then, since intragranular misorientations are strong, it can induce another bias depending on the selected point inside the grain... I understand it in order to compare to "old" texture measurements that were done manually, but since the info is included into the "full" texture, not sure it is really interesting...

The one point per grain (OPPG) stereonets are based on the average orientation of each grain, so the results are not biased by intragranular misorientations. The Reviewer is correct about OPPG stereonets not including information on grain size; they magnify the contributions of finer grains to the CPOs. Differences between OPPG and all data plots may relate to different CPOs in different grain size fractions. The primary reason we show OPPG stereonets (as pointed out by the Reviewer) is that these allow a better comparison to older manual measurements of c-axis orientations (until recently, for all published data). Since those OPPG stereonets do not take up a lot of space in the figures, we have included them for the reasons stated.

6. In paragraph 2.5, about grain size measurement. How does it take into account the grain shape anisotropy? Such a method seems to be well adapted for a microstructure that evolves in staying "self-similar", but it is not the case at all, since the shape anisotropy (and the distribution of grain size) evolves with strain. Maybe you could just mention such a limitation? In the same idea, there is no mention of the fact that you are using a 2D technique to observe and evaluate grains that have a true 3D structures (with anisotropy). We all do that, of course, but the impact is different when a

microstructure is relatively equiaxed, and remain so, or not. In particular, by doing so, we are totally unable to distinguish a small new grain from a cut piece of a highly serrated large grain... It therefore makes it complicated to distinguish nuclei (cf l. 13 of p.8). Again, you are using a "mean grain size", in the case of a non normal distribution, this mean has a weak meaning. Wouldn't a median + quartile representation be more adapted, in order, for instance to follow the evolution of the grain size distribution with strain?

Our grain size measurement does not take grain shape into account. By using an equivalent diameter as the grain size, we minimized the influence of grain shape on the measurement of grain size. This is fairly standard practice in the microstructural community (Berger et al., 2011; Cross et al., 2017; Heilbronner and Kilian, 2017). Grains are not strongly shaped so we do not think that ignoring shape factors will affect measured distributions of grain size significantly.

The Reviewer is correct that this is a 2D technique. As the Reviewer suggested, we have added a sentence on the limitation of this grain size measurement in the Methods section: "Note that grain size determined this way represents the size of a 2D cross section of a 3D grain."

Most recrystallized grain size distributions are skewed (log(d) tends to be normally distributed). We agree that for such non-normal distributions, the mean is not a good scalar representation of the distribution. Therefore, we also presented the "peak grain size" based on the grain size distribution in Figures 5, 6 and 7. Many other studies, however, present the mean — so the mean values here give a comparison for those studies. There is a particular issue in the "ice world" in that many grain sizes are calculated as the mean area (by counting the number of grains in a given area of a thin section). The grain area distribution is highly skewed. More important than the scalar representations of grain size are the measured distributions presented in figures 5, 6 and 7.

7. You refer to SGB, that you assume to be numerous and to evolve with strain. Honestly, this is hard to evaluate from the only figures 5 to 7, since there is no quantitative analyses of it. Relatively to some observations that I have done, I am even surprise not to see more of them, and I would be enable to say that there is a clear evolution with time. Couldn't you estimate, for instance, the Kernel Average Misorientation, and its evolution with time? (cf l.16 p 13 for instance).

We did not assume that the number of subgrain boundaries evolves with strain (and do not say so in the paper). We have not presented data that show quantitatively the subgrain boundary density. In this paper we are trying to focus on the CPO and the minimal microstructural observations required to understand the CPOs. In the Results section, we simply described our observations: subgrain boundaries were observed, and subgrains (cells with at least one low-angle boundary segment) are of similar size to recrystallised grains (cells entirely surrounded by high-angle boundary segments). Subgrain boundaries (grey lines) can be seen in all of the EBSD maps in Figures 5, 6, and 7.

The Reviewer is correct, we do not have a quantitative measurement on the number or density of subgrain boundaries. As suggested, we have added the averaged kernel average misorientation (KAM) for each of the sections in Figures 5, 6 and 7. Averaged KAM provides a measurement of intragranular distortion. The values of averaged KAM do not show a trend with changing strain or temperature. It is not clear whether we should see any increase in KAM across the range of strains explored (between shear strains of 0.5 and 2.5). The presence of low-angle (subgrain) boundaries is indicative of the operation of recovery processes. The decrease in grain size compared to the original grain size requires a nucleation process. The observation that cells with at least one subgrain boundary have similar sizes to grains surrounded entirely by high-angle grain boundaries is consistent with subgrain rotation recrystallization being a nucleation process. These processes can operate within an approximately steady-state microstructure. We might expect an increase in KAM in the first few percent of

strain, but the data shown here are for much larger strains. Clearly, investigating and quantifying internal distortions of grains is an important thing to do, but beyond the scope of this paper. In the paper, we simply present microstructural evidence that is consistent with the operation of recovery and subgrain rotation. Our main focus is to present the CPO patterns and to establish a testable explanation.

8. part 4.3 About the ' angle, well, I have quite a lot to say (sorry...). First, does it really make sense to compare an evaluation of this angle performed based on very different materials, from experiments with very different conditions? In particular, if I want to separate the two population of M1 and M2 orientation, I aim to take into account not only the visible angle between the 2 poles, but also the number of orientations on each pole (some kind of "weighted angle"). Why? Because in some cases there remain so few orientations in the M2 pole, that the separation is doubtful (and strongly dependent on the measurement technique). For instance, my interpretation of the gamma = 2 texture in BD82 is that there is no more M2 (or too few to be considered, and this was also the interpretation of BD82). Therefore, in this case, the phi angle has not meaning. I would do the same interpretation on your figure 2d at -20 and -30C. Then, from my point of view, figure 8 is highly confusing. (a) you compare experiments made in drastically different conditions. For instance Burg et al. worked on thin plates of ice, with very few grains, and strong boundary conditions effects are expected. Kamb added some axial compression at different levels, Hudleston samples are from natural fault... (b) the weakness of the model used by Llorens et al. is also hidden in the figure. This model, by construction (because it requires non basal prismatic and pyramidal slip system to accommodate the deformation) is unable to stabilise the M1 pole to the vertical, as it is observed naturally or in the laboratory. Although the angle can be small between this pole and the vertical, it persists and the main reason is a non adapted representation of the mechanisms that are accommodating basal slip. Even when adding some representation of subgrain mechanisms can't they correct this bias. But when plotting only the phi angle, this problem is hidden, and the interpretation can be biased... Well, I am not sure that this phi angle evolution is so necessary to the

interpretation of your results, and maybe this one would be clearer without trying to fit all other existing data???

It is true that focusing on the single angle phi is a simplification. However for the experimental data and the one data set on natural ice we can plot on here (Hudleston, 1977), the primary maximum is, within error, normal to the shear plane, so the angle between the two maxima becomes a useful parameter. Using a simple parameter phi makes it easier for readers to understand our paper, which is the same approach used in Kamb (1972) and Bouchez and Duval (1982). It is directly analogous to measuring the opening angle in open cone (small circle) CPOs from axial deformation experiments (e.g., in Qi et al. (2017)). Understanding the relationship of these angles to strain and deformation conditions might lead to a better understanding of the underlying physical processes.

The Reviewer is correct to state that there are some difficulties, some of which are already stated in the text. For example, we state: "The most likely explanation is that the data in Fig. 8 represent experiments with subtly different kinematics (deviations from perfect simple shear) and contains data from experiments conducted across a range of strain rates (or stresses)." We agree that it is rather too problematic to include the see-through experiments of Burg et al. (1986); these data have numerous attributes that do not match other experiments and undoubtedly relate to the very specific boundary conditions of those experiments. We have removed the Burg et al. (1986) data from the figure and related discussion from the text. All of the experimental data and those from natural samples remaining on the plot are dominated by simple shear deformation (the Kamb data we use are those from simple shear-dominated experiments, as are the Wilson and Peternell data). In reality, we need sets of experiments for constant temperature and strain rate (or stress) where only the shear strain varies to verify the pattern of this plot. In the meantime, we believe it is useful to show the plot as is.

Numerical models provide a tool that allows for scaling to slower (natural) strain rates, which can be incorporated into larger scale ice sheet models. For this reason, the

comparison of laboratory experiments with model outputs is important. We can use the differences between model outputs and observations of natural ice to discover what is missing or incorrect in numerical codes, or what might be a result of a boundary condition in an experiment. Importantly, we can also use such comparisons to design better experiments. Picking values of the phi parameter from models is problematic, because, as the Reviewer states, the main cluster of c axes is oblique to the shear plane normal. We have tried to be as clear as possible on this point in the text. Nevertheless, we can extract two clusters from these models, and the evolution of the angle between them provides insight into what processes might be going on in the samples.

9. Part 4.5 Here, again is evoked the increase of subgrain boundary and lattice rotation effect without it being really quantified... It is therefore not so straighforward to assess an evolution from GBM to lattice rotation dominating process during recrystallization.

Here we reiterate that we do not state that subgrain boundaries become more prevalent with strain and the KAM data we have now included support this. Our focus is on the balance between lattice rotation and GBM as a way to explain the CPO evolution. In this context, we state that rotation is controlled by a set of processes including dislocation glide, recovery, subgrain rotation, and grain boundary sliding. We accept that unpacking the details of this and quantifying this balance is rather difficult. Again, this is another aspect for which modeling approaches are important.

10. Part 4.6 This paragraph is, to my point of view, giving a quite simplistic explanation for CPO development. It has been shown in ice and other materials, for quite some time now, that stress and strain field heterogeneities prevent from making a clear distinction between grains "well oriented" and grain "badly oriented" (see e.g. Grennerat et al. 2012, Piazolo et al. 2015 for ice). Such a clear separation is holding when dealing with mean-field modeling approachs that are considering grain as an inclusion in a homogeneous equivalent medium. Full-field modeling approaches such as the one of Llorens et al. does reproduce the complexity. Furthermore, other mechanisms such as twinning (not in ice) and kink-banding (in ice) can be invoked to accommodate basal

slip without the requirement of glide in non-favorable slip systems. And internal distortion does not always need non-basal dislocations to be formed... (tilt bands in ice are made of edge basal dislocations). And internal distortion is not a proxy of the dislocation density! It is only a proxy of the geometrically necessary dislocations, which are not necessarily correlated with the full dislocation density... Maybe some quantitative observation of a relation like subgrain density = f(schmid factor) could help? But we have tried, and we find no relation, such as the results of Grennerat et al. (2012).

Our explanation is deliberately simplistic as we want to present a broad hypothesis that can be tested further. Our approach follows that of Alley (1992) and is influenced by a much larger set of experiments in which samples were deformed via axial shortening (the data in Qi et al. (2017) comprises a small part of this data set), wherein low temperatures and/or high strain rates yield clustering of c-axes parallel to shortening, and high temperatures and/or low strain rates yield an open cone fabric (small circle). In axial shortening, rotation yields the cluster fabric, and the open cone comprises grains with high resolved shear stresses. So, the rationale for assigning the possible relative importance of rotation and GBM to the observed CPOs is clear. Simple shear is more difficult to treat in this way, as the outcome of rotation (to c axes perpendicular to shear plane) coincides with one of the orientations of high resolved shear stress. In axial experiments we would predict an evolution of cone angles with strain. As yet there are no data on cone angles as a function of strain, except for experiments conducted for high temperatures and low strain rates (Jacka and Maccagnan, 1984; Montagnat et al., 2015; Piazolo et al., 2013; Vaughan et al., 2017), wherein the cone angle remains approximately constant as a function of strain. We have now conducted lower-temperature axial compression experiments to increasing strains, but have not yet acquired the microstructural data.

We agree with all of the Reviewer's comments related to kink bands in ice. It is notable that we do not observe kink structures in these shear experiments. In contrast, kinks are common in axial compression experiments at similar temperatures and strain

rates to ours, and in shear experiments at lower temperatures (equivalent to the -20C and -30C experiments here) (Craw, 2018; Seidemann et al., 2018) and the kinking influences the recrystallization behavior (Seidemann et al., 2018). We agree that internal distortion is not a proxy for dislocation density. We have removed "(a proxy for dislocation density)" in section 4.6 page 13.

11. Part 4.7 seems highly speculative to me. Mainly because this could very likely result from this specific experimental set-up on which a compression or tension component may add to the shear deformation (see for comparison results of Duval 1981, JOG vol 27 who shows the effect of adding some compression on a shear experiment. Although they do not measure a-axis orientation, their resulting c-axis orientations could explain partly your a-axis distributions). Just to let you know, we do not observe this specific distribution in our recent torsion tests on ice (Journaux et al. AGU 2017). On more point on this paragraph, related to the references given for c-axis clusters observe in nature: it seems to me more fair to cite pioneer works when they exist (at least some) than to always refer to the review work or the more recent work.

This is the first-time that a-axes have been measured in experimentally sheared ice samples. In our experiments, for large strains, the a axes are parallel to the shear direction. If this were generally true it has important implications for the interpretation of CPOs of naturally deformed ice: a axes could be used to suggest shear directions. We therefore think it is important to include this section.

We thank the Reviewer for sharing their recent experimental observations. Besides the difference in deformation kinematics - ours were simple shear with a small component of compression, and the Reviewer's were (we think, from the description) perfect simple shear — other conditions may affect the distribution of a-axes. For example, in our high temperature (-5°C) experiments, a axes do not align parallel to shear until high strains. The Reviewer mentioned that her experiments were conducted at high temperatures. We guess, because the experiments were at atmospheric pressure, that stresses (and thus strain rates) will be necessarily lower than in our experiments, which has a similar

effect to being at higher temperature (Hirth and Tullis, 1992; Qi et al., 2017; Tullis et al., 1973). It is possible that temperature, and its effect on deformation and recrystallization processes, is a key control on the distributions of a axes. To solve this problem, we believe it is important to publish the a-axis distributions we have observed, and encourage others, including the Reviewer's group, to do the same. We added citations for Gow and Williamson (1970), Herron and Langway (1982) and Herron et al. (1985) to the first sentence of this subsection.

12. Conclusion: From the remark about paragraph 4.7, I would suggest not to mention the point 4 of the conclusion, since it has not been shown that the observed results are not related to the specific experimental conditions used here. To turn this observation into a generic tool to interpret natural texture appears to me a little too fast...

We did not include any statements on the interpretation of natural CPO using this observed a-axes clustering. This observation is new and will be worth testing by other researchers. So we think it is important to keep it here.

13. Similar remarks would hold also for the point 5, and the mentioning of the elongation of c-axis cluster that has not been observed in previous work (for instance Bouchez and Duval 1982).

The elongation of the c –axis clusters is very common in shear experiments and is present in all of our results. Irrespective of the explanation, this is a factual observation and like the a-axis clustering should be reported as a conclusion.

14. About point 6, please refer to my comment on subgrain observations that are very weak in this paper. In order to be able to provide some info about subgrain size, one would need a measurement of this size, or at least a proxy, and it is not given in this work.

It is true that we do not provide a measurement of subgrain size. In a simple bulk measurement subgrain sizes will always come out smaller than grain sizes (Trimby et

al., 1998). For the microstructural data presented here, a statistical subgrain size value measured using a low misorientation cut off will be essentially the same as the grain size, but that number will not be particularly meaningful, since subgrains only occur in one out of $\sim 100$ grains. In this case, the qualitative observation that the subgrains are of similar size to recrystallized grains is still useful.

15. Point 7, a similar remark holds here too since, in order to link what the author calls "high Schmid-factor grains" to GBM, one would need to show that there exist a relation between the Schmid factor and the grain size, or grain shape for instance (if grain size or shape is taken as a proxy of GBM). And this is not provided here. And the last sentence appears speculative too, since, to be able to discriminate the nucleation mechanisms, one would need to be able to observe nuclei! But, because of the 2D observation tool used here, one cannot distinguish small new grains from cut part of a serrated grain boundary. On top of that, nucleation by bulging could very well occur, but one would not observe it by only looking at 2D microstructure at the end of the test. Therefore, the only qualitative observation of subgrain boundaries is not enough to discriminate the nucleation mechanism. So this could be your interpretation, or assumption, but to my point of view this is not shown by the results presented here. We agree with the Reviewer that without establishing a relationship between Schmid factor and grain size, it is not a good idea to conclude this way. Therefore, we removed "high Schmid-factor grains", so that the sentences reads "...a balance of the rates of lattice rotation due to dislocation slip and growth of grains by strain-induced GBM."

Distinguishing nucleation mechanisms is always difficult. We agree that ruling out bulging is very difficult and we have removed the implication that we can rule it out in the text.

Alley, R. B., 1992, Flow-Law Hypotheses for Ice-Sheet Modeling: Journal of Glaciology, v. 38, no. 129, p. 245-256. Becroft, L., 2015, New grain growth experiments in water ice [MSc: University of Otago, 132 p. Berger, A., Herwegh, M., Schwarz, J. O., and Putlitz, B., 2011, Quantitative analysis of crystal/grain sizes and their distributions in

2D and 3D: Journal of Structural Geology, v. 33, no. 12, p. 1751-1763. Bouchez, J. L., and Duval, P., 1982, The Fabric of Polycrystalline Ice Deformed in Simple Shear - Experiments in Torsion, Natural Deformation and Geometrical Interpretation: Textures and Microstructures, v. 5, no. 3, p. 171-190. Burg, J. P., Wilson, C. J. L., and Mitchell, J. C., 1986, Dynamic Recrystallization and Fabric Development during the Simple Shear Deformation of Ice: Journal of Structural Geology, v. 8, no. 8, p. 857-870. Cole, D. M., 1996, Observations of pressure effects on the creep of ice single crystals: Journal of Glaciology, v. 42, no. 140, p. 169-175. Craw, L., 2018, Causes and consequences of heterogenous flow behavior in ice [MSc: University of Otago, 109 p. Cross, A. J., Prior, D. J., Stipp, M., and Kidder, S., 2017, The recrystallized grain size piezometer for quartz: An EBSD-based calibration: Geophysical Research Letters, v. 44, no. 13, p. 6667-6674. Durham, W. B., Heard, H. C., and Kirby, S. H., 1983, Experimental Deformation of Polycrystalline H2o Ice at High-Pressure and Low-Temperature - Preliminary-Results: Journal of Geophysical Research, v. 88, p. B377-B392. Golding, N., Schulson, E. M., and Renshaw, C. E., 2010, Shear faulting and localized heating in ice: The influence of confinement: Acta Materialia, v. 58, no. 15, p. 5043-5056. -, 2012, Shear localization in ice: Mechanical response and microstructural evolution of P-faulting: Acta Materialia, v. 60, no. 8, p. 3616-3631. Heilbronner, R., and Kilian, R., 2017, The grain size(s) of Black Hills Quartzite deformed in the dislocation creep regime: Solid Earth, v. 8, no. 5, p. 1071-1093. Hidas, K., Tommasi, A., Mainprice, D., Chauve, T., Barou, F., and Montagnat, M., 2017, Microstructural evolution during thermal annealing of ice-Ih: Journal of Structural Geology, v. 99, p. 31-44. Hirth, G., and Tullis, J., 1992, Dislocation Creep Regimes in Quartz Aggregates: Journal of Structural Geology, v. 14, no. 2, p. 145-159. Hudleston, P. J., 1977, Progressive Deformation and Development of Fabric Across Zones of Shear in Glacial Ice, in Saxena, S. K., Bhattacharji, S., Annersten, H., and Stephansson, O., eds., Energetics of Geological Processes: Hans Ramberg on his 60th birthday: Berlin, Heidelberg, Springer Berlin Heidelberg, p. 121-150. Jacka, T. H., and Maccagnan, M., 1984, Ice Crystallographic and Strain Rate Changes with Strain in Compression and Extension:

Cold Regions Science and Technology, v. 8, no. 3, p. 269-286. Jones, S. J., and Chew, H. A. M., 1983, Creep of ice as a function of hydrostatic-pressure: Journal of Physical Chemistry, v. 87, no. 21, p. 4064-4066. Li, J., Jacka, T. H., and Budd, W. F., 2000, Strong single-maximum crystal fabrics developed in ice undergoing shear with unconstrained normal deformation, in Hutter, K., ed., Annals of Glaciology, Vol 30, 2000, Volume 30, p. 88-92. Montagnat, M., Chauve, T., Barou, F., Tommasi, A., Beausir, B., and Frassengeas, C., 2015, Analysis of dynamic recrystallisation of ice from EBSD orientation mapping: Frontiers of Earth Science, v. 3, p. 13. Piazolo, S., Wilson, C. J. L., Luzin, V., Brouzet, C., and Peternell, M., 2013, Dynamics of ice mass deformation: Linking processes to rheology, texture, and microstructure: Geochemistry Geophysics Geosystems, v. 14, no. 10, p. 4185-4194. Qi, C., Goldsby, D. L., and Prior, D. J., 2017, The down-stress transition from cluster to cone fabrics in experimentally deformed ice: Earth and Planetary Science Letters, v. 471, p. 136-147. Rigsby, G. P., 1958, Effect of Hydrostatic Pressure on Velocity of Shear Deformation of Single Ice Crystals: Journal of Glaciology, v. 3, no. 24, p. 271-278. Seidemann, M., Prior, D. J., Golding, N., Durham, W. B., Lilly, K., and Vaughan, M., 2018, The role of kink boundaries in the shear localisation of polycrystalline ice: Journal of Structural Geology, v. In Review. Trimby, P. W., Prior, D. J., and Wheeler, J., 1998, Grain boundary hierarchy development in a quartz mylonite: Journal of Structural Geology, v. 20, no. 7, p. 917-935. Tullis, J., Christie, J. M., and Griggs, D. T., 1973, Microstructures and Preferred Orientations of Experimentally Deformed Quartzites: Geological Society of America Bulletin, v. 84, no. 1, p. 297-314. Vaughan, M., Prior, D. J., Brantut, N., Jefferd, M., Mitchell, T. M., and Seidemann, M., 2017, Insights into anisotropy development and weakening of ice from p-wave velocity monitoring during creep.: Journal of Geophysical Research, v. In Press.

Please also note the supplement to this comment:
https://www.the-cryosphere-discuss.net/tc-2018-140/tc-2018-140-AC1-supplement.pdf

[Figure]

[Figure]

Replotting of Bouchez and Duval 1982 data. All processing in MTEX

[Figure]

**Fig. 1.** Figure R1. Replotting of the orientation data of Bouchez and Duval (1982) with MTEX toolbox.

---

## Author Comment (AC2) · 13 Nov 2018

Response to Reviewer 2

We thank Reviewer 2 for their thoughtful review. We present our detailed responses (shown in black text) to the Reviewer's comments (shown in blue).

1. During the sample preparation, when the samples are cooled to -60°C for the welding of the indium jacket, is there any possibility for the thermal/confinement stresses to alter the microstructure as it would relate to the grown-in dislocation density?

Each experimental sample, including the undeformed one published in Qi et al. (2017),

were stored in liquid nitrogen before microstructural characterization in the SEM. No thermal-stress-related microstructure was observed in these samples. Transferring the sample from the -30°C freezer to the -60°C alcohol bath for welding is a modest temperature change compared to immersing a sample from the freezer in liquid nitrogen. Thus, transferring the sample from the -30°C freezer to -60°C alcohol should not induce any observable alterations to the microstructure.

Moreover, after welding the jacket, each sample was placed into the apparatus and maintained at the temperature and pressure of the deformation experiment for >1h, so that if any dislocations were produced during welding, those dislocations would be relaxed before deformation started.

2. From looking at Figure 1, I am perplexed as to how the piston is able to translate laterally while also remaining rigid and in-line with the axis of compression? Could you please explain?

As illustrated in the figure below, as the driving piston moves vertically up, the bottom 45°-cut piston moves sideways, because the ice sample is weaker than the pistons and the boundary between the assembly and the driving piston allows lateral movement. This design for shear deformation is widely used in rock deformation studies (e.g., Schmid et al., 1987, JSG; Dell'angelo and Tullis, 1989, Tectonophysics). We have modified Figure 1 to include this information.

3. Could you include the data (via personal communication) related to the flow law of the indium jacket and perhaps also the company/supplier that is used? Such that these experiments could be repeated?

We can share W. Durham's indium data with you. But as the data belong to Durham, we think it is not appropriate to publish it in our paper.

4. Please add a citation for Line 1-2, page 5, regarding using the minimum strain rate in creep tests.

We added a citation (Jacka and Maccagnan, 1984) here as suggested.

5. In Section 3.4, please comment on the skewed distribution of grain sizes. Is this lognormal? As would be expected? Was this distribution used for calculating the mean? How was the anisotropy in the elongated grains accounted for?

For most samples, the distribution is lognormal. A lognormal distribution of grain size was expected, because our previous compression experiments found roughly the same distribution. Moreover log normal distributions of dynamically recrystallized grain size in rocks and metals are very common. The distribution was not used to calculate the mean grain size, but was used to identify the "peak" grain size. Using an equivalent radius to calculate grain size, the influence of anisotropic grain shapes is minimized.

6. Discussion Section. Although I appreciate the detail of this section, it seems to me that it could be more concise, such that the most relevant findings and results are more impactful.

We have shortened the discussion section by about half a page.

7. In Section 4.1, should any consideration be given to the recrystallized grains experiencing primary creep in this scenario?

The terms "primary creep", "secondary creep" and "tertiary creep" are used to describe the behavior of the bulk of a material. In tertiary (approximately steady-state) creep, dynamic recrystallization and grain growth are balanced. Recrystallized grains with lower dislocation densities than other grains are produced at all times. But researchers do not generally consider these recrystallized grains to be under primary creep, because the bulk material is in the tertiary creep stage.

8. Page 11, line 30, replace "in" with "are".

Changed as suggested.

9. Line 13-14, Page 13, please add a citation for this statement.

We added a citation (Steinbach et al., 2017) as suggested.

10. Regarding GBS mentioned in Section 4.6, was there any evidence of this in the observed microstructure? Quadruple points? If not, how would this be incorporated into models for ice if it has yet to be directly observed?

GBS is always a tricky process to infer because, unlike dislocation creep, it does not leave clear microstructural signatures, or its characteristic microstructures have not yet been identified. Furthermore it is unlikely that basal-slip accommodated GBS dominates the deformation at the high stresses explored in our experiments, in which dislocation creep is likely also occurring. Our EBSD maps contain numerous examples of quadruple junctions and near-quadruple junctions. Similar observations have been used to infer GBS in other materials (Negrini et al., 2018) but the issue is complex (Kellermann-Slotemaker and De Bresser, 2006) and needs a more systematic investigation to be useful in these experiments. The microstructures of the samples deformed at -20°C and -30°C are very similar to those in the recrystallized portions of samples deformed by Craw et al. (2018). In the Craw et al. work, the (natural) samples had much larger original grains, and explaining the CPOs of the porphyroclasts and recrystallized grains is much, much easier if GBS is allowed. We cannot prove that GBS is occurring in our experiments, but we can infer it is active based on extrapolation of existing flow laws for GBS to the conditions of our experiments, and that it will then influence the evolution of the CPO. Modelling GBS allows us to explore its potential effects on CPO in a more rigorous way. Another indication that GBS is likely is that peak stresses at a given strain rate are grain size-dependent (Qi et al. 2017). Grain size sensitivity, with strain rate increasing with decreasing grain size, requires grain boundary sliding (Langdon, 2006) for kinematic reasons, irrespective of whether sliding is accommodated by diffusional or dislocation processes. As we have evidence that there is grain size sensitivity (and thus a component of GBS) at the (larger) starting grain size (Qi et al. 2017), it is likely that GBS becomes even more important as the grain size is reduced with strain.

11. Conclusions Section. Could also be more concise. (e.g. no need to summarize the method and/or results before presenting a conclusion)

We consolidated this section as suggested. The summary of the experimental methods a was removed.

12. Figure 3b – and with regard to Question 2: : :Am I correct to interpret the increase in the shear stress in these tests as related to the piston becoming displaced and the onset of frictional effects? If not, could you further explain the cause of the increase in the shear stress?

The raw data are illustrated in Figure 3a. The stress is roughly stable with increasing strain in the raw data, which suggests there is no onset of an additional frictional force. In Figure 3b, the increase in the stress is due to the application of the area correction. As shear strain increases, the piston becomes displaced (see Figure R1 above), and the area of the sample that is in contact with both top and bottom pistons decreases.

However, after consulting with experimentalists who are more familiar with the direct shear method (Greg Hirth and Leif Tokle), we decide to remove panel b from Figure 3. They have concluded for a large data set on other earth materials that stress is transferred across the whole cross sectional area up to high strain and that the area correction is not needed. The observation that the grain sizes of our samples do not change much with increasing strain, especially in the -20 and -30°C experiments, suggests that the stresses in our experiments are roughly constant with increasing strain at strains > 0.2. This confirms that an area correction is not necessary.

13. Figures 5,6,7 – It's not clear to me what is being indicated with the blocky black arrow on the left of these maps. Is this a transverse view/map?

These black arrows mark the shear direction. These figures are transverse view, which we called it shear plane, as illustrated in Figure 1. You can see the black arrows in Figures 1 and 2. In the captions of Figures 5, 6 and 7, we mentioned that: "The shear

direction on the top side is up, as shown by the black arrow."

14. Figure 10 – Is it possible to also quantify the Key Processes related to the Final microstructure? Such that these 2-D characteristics could be identified with an automated algorithm? Perhaps see Lehto et al. 2016, Characterization of local grain size variation of welded structural steel, as a good starting point. It seems that there needs to be a better method of identifying and/or quantifying the differences in these microstructural regimes.

Thank you for suggesting Lehto et al. 2016. It is a very interesting paper, but beyond the scope of the discussion in our paper. In our paper, we are focusing on the observed transition in the CPO patterns. We use microstructural evidence to support our explanation of CPO formation, but at the current stage it is very difficult to quantify the contributions from different recrystallization processes. We have considered kernel average misorientation and subgrain boundary density, but neither of them is a good proxy for a recrystallization mechanism. This is an area of future research for our groups.

15. Lastly, after reading Maurine Montagnat's insightful comments pertaining to this manuscript, I would have to agree that it is difficult to ascertain with any certainty the nucleation mechanisms responsible for recrystallization from a 2-D post-mortem analysis alone.

Yes. There are difficulties in determining the nucleation mechanism from a 2D section. We have removed the sentence related to the nucleation process in the conclusion section.

Dell'angelo, L.N. and Tullis, J., 1989. Fabric development in experimentally sheared quartzites. Tectonophysics, 169(1-3), pp.1-21.

Jacka, T.H. and Maccagnan, M., 1984. Ice crystallographic and strain rate changes with strain in compression and extension. Cold Regions Science and Technology, 8(3),

pp.269-286.

Langdon, T.G., 2006. Grain boundary sliding revisited: Developments in sliding over four decades. Journal of Materials Science, 41(3), pp.597-609.

Negrini, M., Smith, S.A., Scott, J.M. and Tarling, M.S., 2018. Microstructural and rheological evolution of calcite mylonites during shear zone thinning: Constraints from the Mount Irene shear zone, Fiordland, New Zealand. Journal of Structural Geology, 106, pp.86-102.

Qi, C., Goldsby, D.L. and Prior, D.J., 2017. The down-stress transition from cluster to cone fabrics in experimentally deformed ice. Earth and Planetary Science Letters, 471, pp.136-147.

Schmid, S.M., Panozzo, R. and Bauer, S., 1987. Simple shear experiments on calcite rocks: rheology and microfabric. Journal of structural Geology, 9(5-6), pp.747-778.

Slotemaker, A.K. and De Bresser, J.H.P., 2006. On the role of grain topology in dynamic grain growth—2D microstructural modeling. Tectonophysics, 427(1-4), pp.73-93.

Steinbach, F., Kuiper, E.J.N., Eichler, J., Bons, P.D., Drury, M.R., Griera, A., Pennock, G.M. and Weikusat, I., 2017. The relevance of grain dissection for grain size reduction in polar ice: insights from numerical models and ice core microstructure analysis. Frontiers in Earth Science, 5, p.66.

Please also note the supplement to this comment:
https://www.the-cryosphere-discuss.net/tc-2018-140/tc-2018-140-AC2-supplement.pdf

———————————————

[Figure]

[Figure]

**Fig. 1.** Figure R1. Schematic drawing illustrating the movement of the pistons during deformation.

[Figure]

**Fig. 2.** Figure R2. Plot of differential stress vs. temperature for indium. Data from W. Durham.

[Figure]

---

## Referee Report (RR1)

Comments on "Crystallographic preferred orientations of ice deformed in direct-shear experiments at low temperatures" by Chao Qi et al.

These comments are made following two initial reviews and submission of the revised MS.

The MS is well written and well organized, and I congratulate the authors on a fine piece of original research. This is an important contribution to the experimental work on ice rheology, with a focus on the mechanisms that accommodate deformation and the crystallographic fabric that results from the operation of those mechanisms, and thus on the rheology that depends on both mechanisms and fabric.

The experimental procedure is well explained, the results clearly presented, and the interpretation and discussion well founded, leading to questions that should be addressed in future work. I have a few specific comments, which may or may not be addressed before final publication.

p. 1, line 2  I don't understand what is meant by "equivalent to an extrusion of 150%" There is no extrusion in these experiments (except by the limited bulging, but that is minor). This figure of 150% relates to the "equivalent strain" shown on Fig. 3, but this is also confusing to me. It requires a definition.  For another possible measure of strain intensity, a simple shear strain of 2.6 - the maximum in these experiments - corresponds to a maximum stretch of 2.94 or an extension of 1.94.

p. 4, line 16   It seems to me the statement here should be the other way round. That is, if gamma measured and gamma calculated are very close in value, it implies that epsilon axial is very small, and thus bulging is slight. If bulging is calculated first, how is it done?

p. 6, line 8-9   "There are almost no subgrain…."

p. 6, line 13-15.  I would interpret flattening strain to mean shortening normal to the shear plane, not axial  shortening, and that this would be determined by the difference between ho and h1 - although I see from Table 1 that the measurement of h1 is not precise.  The "axial strain" as calculated here, as I understand it, would only be a true axial strain if the deformation were coaxial, which it is not.

p. 9, line 11    "There is a range of grain ….."

p. 9, line 16-18     The stress drop following peak stress (Fig. 3) is rapid, yet the possible reasons for this given here - weakening due to grain size reduction and

thus increase in the contribution of grain size sensitive deformation mechanisms, and geometric softening due to the development of CPO are both likely to be gradual.  So why so rapid a drop?

p. 10, line 3-4   The flattening that occurs in the samples is not just accommodated by extension normal to the shear direction but also by extension parallel to the shear direction. So it's not clear why there should just be preferred elongation in the direction normal to the shear direction. Why not a broadening in all directions?

Figs. 5-7.  The fields of view shown may be deceptive, but the average grain size and number of grains counted are hard to reconcile with the images.  In particular, why only 144 grains in the lowest strain sample at -5 and 3000+ grains for the highest strain sample at -30 for a grain size that is on average larger?

Figs. 5-7.  In principle the plane of section here - the shear plane - is a plane of no strain, so one would expect to find little or no SPO. It would be interesting to see sections perpendicular to the shear plane and containing the shear direction, where one would expect to see a more pronounced SPO, and where it might be possible to get more information about the recrystallization mechanisms.

Also, to the extent that there is SPO it should be in the shear direction, yet in the image for the highest strain at -30, the direction of max elongation appears to be at an angle to the shear direction. Is this just an illusion?

---

## Author Response (AR2)

Response to Peter Hudleston,

We thank Peter for his thoughtful review. We present our detailed responses (shown in black text) to the Reviewer's comments (shown in blue).

*p. 1, line 2 I don't understand what is meant by "equivalent to an extrusion of 150%" There is no extrusion in these experiments (except by the limited bulging, but that is minor). This figure of 150% relates to the "equivalent strain" shown on Fig. 3, but this is also confusing to me. It requires a definition. For another possible measure of strain intensity, a simple shear strain of 2.6 - the maximum in these experiments - corresponds to a maximum stretch of 2.94 or an extension of 1.94.*

Apologies- extrusion is not what we meant. The word should have been "extension". And the reviewer is correct, the extension is 1.94 not 1.5. The purpose of the statement in the abstract is to make the magnitude of the strain clearer to readers who are not so familiar with shear strain. We have changed this statement to "….up to shear strain ($\gamma$) = 2.6, equivalent to a maximum stretch of 2.94 (final line length is 2.94 times the original length)."

*p. 4, line 16 It seems to me the statement here should be the other way round. That is, if gamma measured and gamma calculated are very close in value, it implies that epsilon axial is very small, and thus bulging is slight. If bulging is calculated first, how is it done?*

We have some redundancy in our measurements here.  For each sample, we were able to measure the thickness before deformation. For most we have measurements after deformation, although these lack precision. For all but three samples we have a good estimate of shortening perpendicular to the shear plane (see response to your comment later) from the cross sectional area of "bulge" formed by lateral extrusion. We have measurements of piston offset (perpendicular to cylinder axis) and axial piston displacement (LVDT measurements). Piston offsets and axial displacements are similar suggesting that simple shear dominates. Shortening strain perpendicular to the shear plane is small compared to simple shear values calculated from measured offsets, also suggesting that simple shear dominates. We restructured the first paragraph in Section 2.3 to clarify this.

*p. 6, line 8-9 "There are almost no subgrain…."*

Corrected!

*p. 6, line 13-15. I would interpret flattening strain to mean shortening normal to the shear plane, not axial shortening, and that this would be determined by the difference between ho and h1 - although I see from Table 1 that the measurement of h1 is not precise. The "axial strain" as calculated here, as I understand it, would only be a true axial strain if the deformation were coaxial, which it is not.*

This is a good point. Axial strain is not precise. We actually mean flattening strain. We have changed this to flattening strain in the text.

*p. 9, line 11 "There is a range of grain ….."*

Corrected!

*p. 9, line 16-18 The stress drop following peak stress (Fig. 3) is rapid, yet the possible reasons for this given here - weakening due to grain size reduction and thus increase in the contribution of grain size sensitive deformation mechanisms, and geometric softening due to the development of CPO are both likely to be gradual. So why so rapid a drop?*

This comment highlights something we had not thought about. The rate of stress drop appears very rapid because the strain scale goes to higher strains than we are used to looking at. If the figure is plotted on approximately the same scale as an axial experiment, it is clear that the stress drop rate in axial and shear experiments are comparable. Most of the weakening occurs in the first 10% to 15% of strain with the steepest stress-strain slope between peak stress and ~5% strain. The stress drop rate also corresponds approximately to the rate of strain-rate enhancement in constant load creep tests. The figure below plots the direct shear data at approximately the same strain scales as axial constant rate (Qi et al., 2017) and constant load (Budd et al., 2013) experiments. We know from up-strain, warm experiments that CPO development can correlate to the weakening that occurs over the first 10% of strain (Vaughan et al., 2017). Unpublished work shows that grain size distributions can also evolve in a way that correlates with mechanical weakening ((Vaughan, 2016)(and new -10 and -30°C up strain axial experiments we have just analysed in the last three weeks). CPO and grain size both correlate to "strength" and provide plausible explanations for weakening. We still do not understand this in detail. In the text we have clarified this by adding two sentences in Section 4.1: "Because the x-axis in Fig. 3 is scaled to large strains, the decrease in stress with strain after the peak stress seems more dramatic than in axial compression experiments.  In fact, the evolution of stress with strain in our experiments is very comparable to that in axial compression tests (Qi et al., 2017) when data from both types of experiment are plotted at the same scale.  Furthermore the pattern of decrease of stress with strain from peak stress matches the pattern of increase of strain rate with strain in constant load experiments in both axial compression and in shear (Budd et al 2013).

[Figure]

Figure 1. Plots of stress/strain rate vs. strain. All strains are equivalent to axial compressive strains.

*p. 10, line 3-4 The flattening that occurs in the samples is not just accommodated by extension normal to the shear direction but also by extension parallel to the shear direction. So it's not clear why there should just be preferred elongation in the direction normal to the shear direction. Why not a broadening in all directions?*

Li et al. (2000) provided this explanation for the elongation of the c-axis clusters. We agree with the reviewer that flattening also occurs by extension parallel to the shear direction, and it is not clear why the elongation is normal to the shear direction based on the explanation by Li et al. (2000). It should be noted that simple-shear simulations by Llorens et al. (2016 EPSL, 2017 Phil Trans) also show an elongated maximum, even though there is strictly no flattening involved.

To fully explain this phenomenon, further analyses of the CPO development in the numerical simulations is needed.

We added a sentence after the sentence in lines 3-4: "However, the explanation by Li et al. (2000) is not entirely satisfactory, as (i) flattening is in all directions perpendicular to the shear plane and (ii) the elongated maximum is also observed in numerical simple-shear simulations by Llorens et al. (2016, 2017), in which no flattening strain is allowed. In these simulations, elongation is most pronounced at low strain rate where recrystallisation has a stronger effect on the CPO than at high strain rates. First assessment of the numerical CPO development suggests that the elongation may result from the orientation-dependent rotation rate of c-axes towards the steady-state maxima.

*Figs. 5-7. The fields of view shown may be deceptive, but the average grain size and number of grains counted are hard to reconcile with the images. In particular, why only 144 grains in the lowest strain sample at -5 and 3000+ grains for the highest strain sample at -30 for a grain size that is on average larger?*

The images shown do not all include all of the area mapped at that step size. Our design of the figures was to present similar sized areas at the same scale to facilitate easy comparison of microstructures. For some of the maps (e.g. at -30°C) we have mapped much bigger areas than shown in the figure. If we show the whole area, the microstructural detail will be lost. Although the images are (in most cases) a sub-area of the full map, the grain size distributions and KAM are derived from the whole mapped area. We have added a sentence in Section 3.4 on page 8: "Note that except for two −5°C samples (PIL91 and PIL82), the analyses of grain size and KAM are based on larger areas than those presented in the figures."

For the samples deformed at -5°C, large portions of the surfaces were not preserved, so we were not able to analyze large areas. This relates directly to using wooden pistons (used for the first three experiments); removing the sample from wooden pistons without damage was difficult. For the samples deformed at -30°C, we were very successful at preserving and polishing the surfaces, allowing us to map larger areas.

*Figs. 5-7. In principle the plane of section here - the shear plane - is a plane of no strain, so one would expect to find little or no SPO. It would be interesting to see sections perpendicular to the shear plane and containing the shear direction, where one would expect to see a more pronounced SPO, and where it might be possible to get more information about the recrystallization mechanisms. Also, to the extent that there is SPO it should be in the shear direction, yet in the image for the highest strain at -30, the direction of max elongation appears to be at an angle to the shear direction. Is this just an illusion?*

We are very grateful that the reviewer pointed out the issue with the highest strain sample (PIL135) in Figure 7. After re-examining the figure, we found that it was not aligned with the shear direction (the black arrow in the figure). We have now re-aligned the image. The grain shape is not as obvious in the sub-area chosen for our re-aligned image, and we now realize that we were making statements based only on the area shown in the figure, rather than the whole area mapped. Across the whole mapped area, there is little SPO.

Imaging profile plane sections was much more challenging than imaging the shear planes. At the time of drafting of the paper, we had some profile plane data and tried to include it. The resulting manuscript structure was confusing and detracted from the key focus on the CPO. We have subsequently developed the methods to get data from the profile planes (the "edges" of 5 mm thick ice plates) and we now have excellent profile plane data for the -20 and -30°C experiments and some data for the -5 °C experiments. The CPOs from profile plane maps are indistinguishable from the CPOs measured from the shear plane. There is potential for much more detailed analysis of the microstructures (misorientations, weighted mean Burgers vectors, CPOs based on grain size fractions, spatial arrangements of grains contributing to the two clusters, etc.) that is beyond the scope of this paper. A full presentation of methods and results will add significant length and we feel this will detract from the primary objective of presenting the CPO data in a succinct manner. We intend to publish more detailed work on the microstructures and this will include the new profile plane data.